# SceneStreamer: Continuous Scenario Generation as Next Token Group Prediction

**Zhenghao "Mark" Peng, Yuxin Liu, Bolei Zhou**
University of California, Los Angeles

## Abstract

Realistic and interactive traffic simulation is essential for training and evaluating autonomous driving systems. However, most existing data-driven simulation methods rely on static initialization or log-replay data, limiting their ability to model dynamic, long-horizon scenarios with evolving agent populations. We propose SceneStreamer, a unified autoregressive framework for continuous scenario generation that represents the entire scene as a sequence of tokens, including traffic light signals, agent states, and motion vectors, and generates them step by step with a transformer model. This design enables SceneStreamer to continuously introduce and retire agents over an unbounded horizon, supporting realistic long-duration simulation. Experiments demonstrate that SceneStreamer produces realistic, diverse, and adaptive traffic behaviors. Furthermore, reinforcement learning policies trained in SceneStreamer-generated scenarios achieve superior robustness and generalization, validating its utility as a high-fidelity simulation environment for autonomous driving. More information is available at https://vail-ucla.github.io/scenestreamer/.

## 1 Introduction

Simulating realistic and diverse traffic scenarios is vital for the development and evaluation of autonomous driving systems. Simulation enables safe, cost-effective, and repeatable testing of driving policies without relying on real-world deployment. However, most existing frameworks use static traffic generation methods, such as replaying logged trajectories from real-world datasets (Li et al., 2023; Dosovitskiy et al., 2017; Gulino et al., 2023). Although faithful to real driving behaviors, they lack interactivity as background agents do not respond to the ego vehicle's actions, limiting their utility for closed-loop evaluation.

Recently, data-driven generative models have emerged to learn to synthesize traffic scenarios from real data, offering a path toward richer and more realistic simulations (Zhang et al., 2025a; Rowe et al., 2025). Learning-based traffic simulation is commonly framed as a motion prediction problem: given a history of agent states in a scene, including map, signals, and initial state of agents, a policy generates the future trajectories of all agents. However, most such models are trained as a one-shot prediction model (Ngiam et al.; Shi et al., 2022; Pronovost et al., 2023) and do not explicitly model interactions between agents during the prediction horizon, which leads to covariate shift when they are unrolled in the simulation. Small prediction errors can compound, causing the simulator to visit out-of-distribution states and produce unrealistic outcomes. Recently, autoregressive models have been proposed to better fit into the driving behavior modeling, especially in the context of closed-loop simulation (Suo et al., 2021; Zhang et al., 2023b; Seff et al., 2023; Kamenev et al., 2022). However, these models still rely on the provided initial states of agents and miss the diversity that emerges from the initial layout of traffic participants. Some other works propose generating the initial conditions and then conducting motion prediction based on these conditions (Feng et al., 2023; Bergamini et al., 2021; Tan et al., 2021). This separation can be inefficient and inflexible, as it prevents the model from sharing context between the initialization and motion prediction phases. It also means the number of agents is fixed at initialization, disallowing new traffic participants to enter the scene over time. But in reality, traffic is an open system: new participants continuously enter the scene while old ones leave (e.g., vehicles turning into or from a side street).

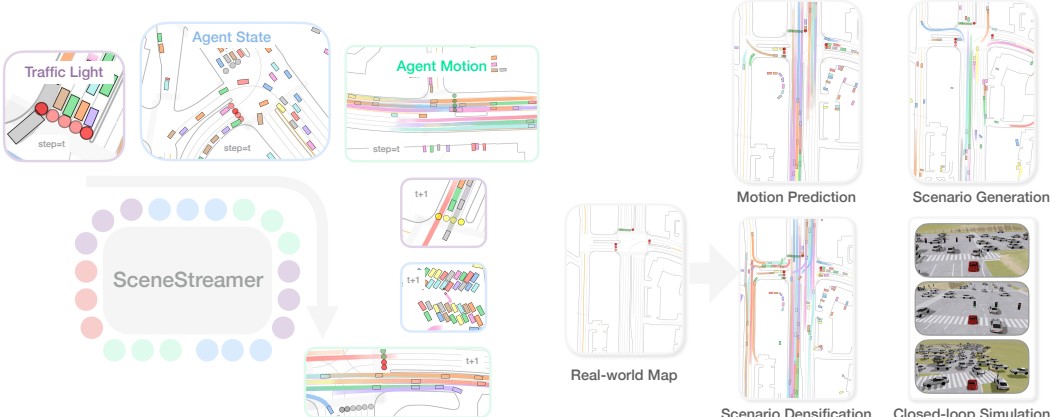

Figure 1: **SceneStreamer enables unified scenario generation via autoregressive token prediction.** We represent a dynamic driving scene using a structured sequence of discrete tokens grouped into traffic light, agent state, and agent motion tokens. SceneStreamer generates these tokens step-by-step on top of static map tokens, allowing flexible and fine-grained simulation. Our unified model supports diverse downstream applications: motion prediction, full-scenario generation from scratch, scenario densification by injecting new agents, and closed-loop simulation for training self-driving planners.

To address these gaps and enable continuous scenario generation, we propose *SceneStreamer*[1], a generative traffic simulation framework that models the entire scenario, including agent states and motions, as a single sequence of tokens. SceneStreamer uses a unified autoregressive model to generate both the agent state and agent motion at every step. We tokenize different categories of agents, such as vehicles, pedestrians, and cyclists, in the same way, with different category embeddings added to their tokens. Notably, SceneStreamer is flexible and can adapt to different tasks, including motion prediction, state initialization, scenario generation, and scene editing (e.g., adding new agents and densification), by choosing different tokens to be state-forced[2] while others are sampled. We implement a carefully designed agent state tokenization pipeline so that the model can effectively handle heterogeneous agent types and map context when adding new agents. Finally, we demonstrate that using SceneStreamer to generate training scenarios leads to significant improvements in downstream planner performance. Reinforcement-learning-based planners trained on SceneStreamer-generated scenarios exhibit greater robustness and better generalization to novel environments. We summarize our contributions below:

1) **Unified State & Trajectory Tokenization:** SceneStreamer employs a single autoregressive model that produces both agents' initial states and their motion trajectories as part of one continuous token sequence over long horizons. This unified approach ensures consistent conditioning between where an agent starts and how it moves, addressing the inflexibility of prior two-stage models.

2) **Agent State Autoregressive Generation:** We design a novel generation scheme for agent states by autoregressively rolling out the agent state tokens and generating the map-based relative states of agents, i.e., the agent's type, its map location, and its detailed kinematic state. This allows the model to accurately place agents on specific map segments (e.g., lanes) and generate realistic state details (position, heading, velocity, etc.) in a compact, learnable representation.

3) **Versatile Capabilities:** By dynamically state-forcing different token groups, SceneStreamer is versatile and applicable to various tasks, including motion prediction, traffic simulation, scenario generation, and scene editing. We demonstrate that training autonomous-driving planners in SceneStreamer-generated scenarios yields more robust and generalizable policies, indicating that SceneStreamer can serve as both a scenario generator and an effective data augmentation tool for closed-loop simulation.

---

[1]An earlier version of this work used the name "InfGen". To avoid confusion with the concurrent work (Yang et al., 2025), we refer to our method as SceneStreamer throughout this paper.

[2]In this paper, we use the term "state-force" to describe directly feeding reconstructed tokens (e.g., agent states at the current step) back into the model, bypassing the generative process. This is distinct from the conventional "teacher forcing" used in sequence modeling, where ground-truth tokens are fed.

## 2 METHOD

**Scenario Generation.** A driving scenario comprises (1) static map context $\mathcal{M}$ (vectorized lane segments, crosswalks, etc.) and (2) dynamic entities including traffic-lights $\{l_t^{(k)}\}_{k=1}^{N_{\text{TL}}}$ and traffic agents $\{a_t^{(i)}\}_{i=1}^{N_t}$ that evolve with time $t$. Here $N_{\text{TL}}$ denotes the number of traffic lights and $N_t$ denotes the number of agents at step $t$. Each traffic-light state $l_t^{(k)} = (x, y, s)$ contains 2-D position and discrete signal $s \in \{\text{green}, \text{yellow}, \text{red}, \text{unknown}\}$ while each agent state $a_t^{(i)} = (x, y, v_x, v_y, \psi, c, l, w, h)$ encodes pose, velocity, heading, category $c \in \{\text{vehicle}, \text{pedestrian}, \text{cyclist}\}$ and length, width and height of the 3D bounding box of the agent. Compared to conventional motion prediction task, which assumes a fixed agent set $\mathcal{I} = \{1, \ldots, N\}$ and access to agent history $\{a_{1:t}^{(i)}\}_{i \in \mathcal{I}}$, and forecasts future trajectories $\{a_{t+1:T}^{(i)}\}_{i \in \mathcal{I}}$, *scenario generation* must create the initial agent set and continually inject new agents, traffic-light changes, and motions over a horizon $T$: $p_\theta(\mathcal{S}_1, \ldots, \mathcal{S}_T | \mathcal{M})$ with $\mathcal{S}_t = (\{l_t^{(k)}\}, \{a_t^{(i)}\})$.

### 2.1 SCENARIO AS A TOKEN SEQUENCE

We cast scenario generation as a next-token prediction task: map tokens $<\text{MAP}>$ are followed *each step* by traffic-light tokens $<\text{TL}>$, agent-state tokens $<\text{AS}>$, and agent-motion tokens $<\text{MO}>$ and form a single autoregressive token sequence: $\mathbf{x}_{1:T} = [<\text{MAP}>; (<\text{TL}>, <\text{AS}>, <\text{MO}>)_1; (<\text{TL}>, <\text{AS}>, <\text{MO}>)_2; \ldots]$. Given all tokens $\mathbf{x}_{<t}$ generated so far, the model predicts the next token or next set of tokens $p_\theta(x_t \mid \mathbf{x}_{<t})$ and samples $x_t$. Following this idea, we develop **SceneStreamer**, a unified transformer that sees the whole history and rolls out the scenario step-by-step, enabling fine-grained, closed-loop generation and smoother downstream simulation integration.

**Map Tokens $<\text{MAP}>$.** A map segment (e.g., road line, stop sign, crosswalk) is represented as a polyline of up to $N_p$ ordered 2D points with semantic attributes. We denote all $M$ segments as $\mathbf{S}_{\text{map}} \in \mathbb{R}^{M \times N_p \times C}$, where $C$ is the per-point feature dimension (e.g., position, road type). A PointNet-like encoder (Qi et al., 2017) yields features $\{\mathbf{m}_i\}_{i=1}^M$, which are passed through the *SceneStreamer Encoder* to produce the map features: $\mathbf{m}' = \text{SceneStreamerEnc}(\mathbf{m})$. To support cross-attention in the decoder, we assign each map segment a unique discrete index $i$ (its map ID), and embed it into the map token:

$$<\text{MAP}>_i = \mathbf{m}'[i] + \text{EmbMapID}(i) \otimes \mathbf{g}_i, \quad i = 1, \ldots, M. \tag{1}$$

where EmbMapID is a learned embedding table. $\otimes$ denotes we will record the geometric information $\mathbf{g}_i$ of map segment $i$, which includes its center position and heading, and use it to participate the relative attention. We defer the discussion of relative attention to Sec. 2.3. The map ID will be used to refer a map segment during agent state generation (Sec. 2.2). These map tokens $\{<\text{MAP}>_i\}$ are provided by the SceneStreamer Encoder and are kept fixed during simulation, serving as static cross-attention keys/values for all decoder layers.

**Traffic Light Tokens $<\text{TL}>$.** Each traffic light is represented by a single token per step. We encode the traffic light's discrete state (green, yellow, red, or unknown), its identifier, and the ID $\lambda_k$ of the map segment it resides in and construct the traffic light token for light $k$ at step $t$ as:

$$<\text{TL}>_{k,t} = \text{EmbState}(s_{k,t}) + \text{EmbTLID}(k) + \text{EmbMapID}(\lambda_k) \otimes \mathbf{g}_k, \quad k = 1, ..., N_{\text{TL}}, \tag{2}$$

where $s_{k,t} \in \{\text{G}, \text{Y}, \text{R}, \text{U}\}$ is the signal state, $\lambda_k$ is the discrete map segment ID the light is attached to, and $\mathbf{g}_k$ is its temporal-geometric context (position, orientation and current timestep). As with map tokens, $\otimes$ indicates that $\mathbf{g}_k$ participates in relative attention (Sec. 2.3).

**Agent State Tokens $<\text{AS}>$.** For every active agent, including newly injected agents at step $t$, SceneStreamer uses a set of four agent state tokens that collectively encode the agent's dynamic and semantic state. These states include positions, headings, velocities, shapes and agent categories. As shown in Fig. 3(A), each agent $i$ present at step $t$ is represented by four ordered tokens: $\langle <\text{SOA}>_i, <\text{TYPE}>_i, <\text{MS}>_i, <\text{RS}>_i \rangle_t$. Here $<\text{SOA}>$ is the start-of-agent flag, $<\text{TYPE}>$ is the categorical token in $\{\text{vehicle}, \text{pedestrian}, \text{cyclist}\}$, $<\text{MS}>$ is the index of the map segment where the agent resides as shown in Fig. 3(B), $<\text{RS}>$ is the relative states of agent w.r.t. the selected map segment. We defer the detailed composition of each token to the Appendix.

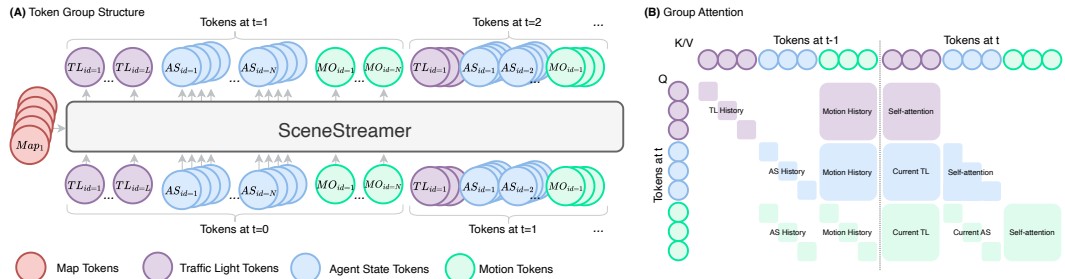

Figure 2: **The tokenization and attention mechanism of SceneStreamer.** **(A)** SceneStreamer autoregressively generates a sequence of tokens representing a full traffic scenario. Each simulation step consists of traffic light tokens (purple), agent state tokens (blue), and motion tokens (green), conditioned on static map tokens (red). This structured tokenization enables step-wise rollout of the dynamic scene and allows new agents to be introduced at any timestep. **(B)** Grouped causal attention governs how tokens interact: each token attends densely within its group and to logically preceding groups, while also incorporating cross-timestep context (e.g., agents attend to their own history). This attention design encodes semantic causality (e.g., agent motion depends on agent state, which depends on map), enabling fine-grained closed-loop simulation with coherent agent behaviors.

The most interesting token is the relative state token. Specifically, relative state $\mathbf{r}_i$ is a 8D vector, each dimension representing a field in the agent's relative state vector: $\mathbf{r}_i = (l, w, h, u, v, \delta\psi, v_x, v_y)$, where $(l, w, h)$ is the agent's physical dimensions (length, width, height), $(u, v)$ is the longitudinal and lateral offset from the centerline of map segment $\lambda_i$, $\delta\psi$ is the heading residual relative to the map segment's orientation, $(v_x, v_y)$ is the velocity vector whose direction is in the frame of the map segment. The relative states $\mathbf{r}_i$ can be autoregressively generated by the relative state head as shown in Fig. 3(C) (see Sec. 2.2). Representing agent states relative to local map segments allows for a unified and compact token vocabulary, avoiding the need to discretize the entire map globally and enabling scalable scenario modeling.

**Motion Tokens $<$MO$>$.** To model agent motion, SceneStreamer predicts a motion label for each agent, parameterized as a pair of acceleration and yaw rate $(a, \omega)$. Given an agent's current state, the next state is predicted using a first-order bicycle model. We bucketize the acceleration and yaw rate space into uniform bins. To obtain the ground-truth (GT) motion token, we enumerate all candidate motion labels $(a, \omega)$ combinations and search for the best candidate with least *Average Corner Error (ACE)*, the mean error between the 2D bounding boxes of a candidate and GT. This strategy ensures that both position and heading are tightly aligned with GT and mitigate the compounding error in the tokenization of GT trajectory.

For an agent $i$ at a timestep $t$, we use a motion token $<$MO$>$ to encode the motion label and the identity-related context. The motion token is computed as:

$$<\text{MO}>_{i,t} = \text{EmbMotion}(\mu_{i,t}) + \text{EmbType}(c_i) + \text{EmbAID}(i) + \text{EmbVel}(\mathbf{v}_i) + \text{EmbShape}(\mathbf{s}_i) \otimes \mathbf{g}_i \tag{3}$$

where $\mu_{i,t}$ is the motion label, $c_i$ is the type of agent, $i$ is agent's ID, $\mathbf{v}_i$ is a 2D vector representing the agent velocity in local frame, and $\mathbf{s}_i$ is a 3D vector of agent's shape (length, width, height). $\mathbf{g}_i$ is a 4D vector encoding the temporal-geometric information (agent's current global position, heading and time step). Note that the embedding tables EmbType and EmbAID are shared with $<$AS$>$.

## 2.2 AUTOREGRESSIVE SCENARIO GENERATION

SceneStreamer is an encoder-decoder model. The SceneStreamer Encoder processes the information of map segments and output $\{<$MAP$>_i\}$. The SceneStreamer Decoder, denoted by SceneStreamerDec, autoregressively generates tokens in a step-by-step manner. As demonstrated in Fig. 2(A), in each step, SceneStreamer first generates a set of traffic light tokens $<$TL$>$ predicting the next state of traffic light signals, then it generates the agent state tokens $<$AS$>$ one-by-one. Finally, the motion tokens $<$MO$>$ of all agents are generated.

**Traffic Light Tokens $<$TL$>$.** All traffic light tokens are generated in a single batch at each step, enabling the self-attention between nearby traffic lights. The output is obtained from the traffic light

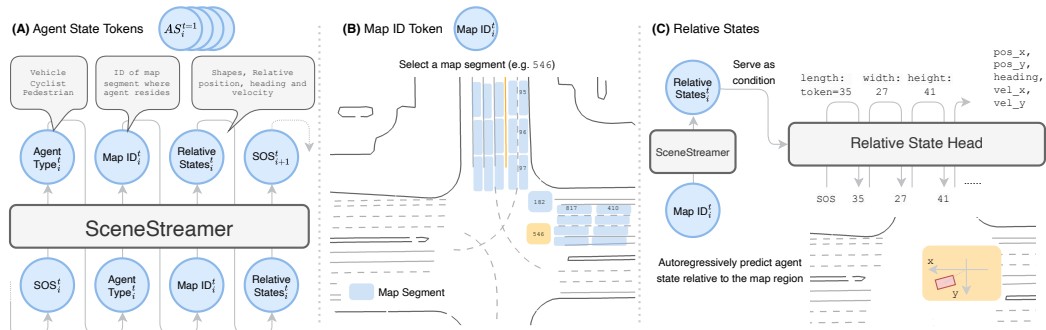

Figure 3: **The design of agent state generation.** **(A)** Each agent's state is encoded as 4 tokens. We first predict the agent type, select a map ID where the agent resides, then predict the relative states. **(B)** Before obtaining the agent state, we first select a map segment as the "anchor" where the agent should reside. **(C)** Feeding in the Map ID, we use the output token as the condition and call the Relative State Head, which is a tiny transformer, to autoregressively generate the relative agent states, including shape, position, heading and velocity.

head, a MLP layer $\text{HeadTL}(\cdot)$ mapping the decoder output to the probabilities of four discrete states: $\{\texttt{green}, \texttt{yellow}, \texttt{red}, \texttt{unknown}\}$:

$$\{s_{k,t}\}_k \sim \text{HeadTL}(\text{SceneStreamerDec}(\{<\texttt{TL}>_{k,t-1}\}_{k=1}^{N_{\text{TL}}})) \in \mathbb{R}^{N_{\text{TL}} \times 4}. \quad (4)$$

**Agent State Tokens $<$AS$>$.** As shown in Fig. 3(A), for one agent, there are four tokens used to represent the agent state. In the test time, we will first sample an agent type from the distribution produced by the agent type head: $c \sim \text{HeadType}(\text{SceneStreamerDec}(<\texttt{SOA}>))$. Then, as shown in Fig. 3(B), the model will select one of the map segment $\lambda_i$: $\lambda_i \sim \text{HeadMapID}(\text{SceneStreamerDec}(<\texttt{TYPE}>))$. After selecting a map segment $\lambda_i$ and generating the associated map segment $<\texttt{MS}>$ token, we condition on the decoder output of $<\texttt{MS}>$ to generate the agent's full kinematic and shape attributes. As illustrated in Fig. 3(C), a dedicated module called the *Relative State Head*, a small Transformer decoder with AdaLN (Perez et al., 2018), is used to autoregressively generate a sequence of 8 tokens, each representing a field in the relative state vector, conditioned by the latent vector from the decoder:

$$(l, w, h, u, v, \delta\psi, v_x, v_y) \sim \text{HeadRS}(<\texttt{SOS}>|\text{SceneStreamerDec}(<\texttt{MS}>)). \quad (5)$$

Here, $(l, w, h)$ is the agent's length, width, height, $(u, v)$ is the longitudinal and lateral offset from the center of map segment $\lambda_i$, $\delta\psi$ is the relative heading to the map segment's orientation, $(v_x, v_y)$ is the relative velocity vector. $<\texttt{SOS}>$ is the start-of-sequence indicator. We bucketize the continuous features $l, w, h, u, v, \delta\psi, v_x, v_y$ so the transformer can predict categorical distributions on them.

For existing agents that persist from the previous timestep, SceneStreamer bypasses the relative state prediction head and instead deterministically state-forces their agent state token using the map segment and relative states. Here, **state-forcing** denotes replacing predicted tokens with reconstructed state tokens whenever the agent's current state is already known. Note that state-forcing will not introduce information leak in inference as we don't read ground-truth agent state from data. This allows SceneStreamer to seamlessly unify dynamic agent injection (via sampling) and agent motion continuation (via state-forcing), ensuring closed-loop autoregressive simulation across variable-length agent sets. Unlike prior methods such as TrafficGen (Feng et al., 2023), which generate all agent state attributes simultaneously in a flat and unstructured output head, SceneStreamer decomposes the generation into a causally constrained sequence and thus can better ensure semantic and physical consistency.

**Motion Tokens $<$MO$>$.** The motion head predicts each agent's motion label as a single categorical token from a 2D discretized space of acceleration and yaw rate. Specifically, we define a flat vocabulary, where each token corresponds to a unique pair $(a, \omega)$ drawn from uniformly quantized grids $\mathcal{A}$ and $\Omega$. A motion prediction head is used to obtain the probability distribution over motion labels. At inference time, we apply top-$p$ (nucleus) sampling to select the motion labels while all motion labels at a step are generated in a single batch:

$$\{\mu_{i,t}\}_i \sim \text{HeadMotion}(\text{SceneStreamerDec}(\{<\texttt{MO}>_{i,t-1}\}_i)). \quad (6)$$

Motion tokens of all agents are generated in one batch as the traffic light tokens, enabling attention between neighboring agents. The sampled token $\mu_{i,t}$ is then mapped back to its corresponding $(a, \omega)$ pair, and passed through a first-order kinematic update rule to compute the next state (see Appendix). At an agent's first appearance, a special label $\mu_{\text{start}}$ is used to get <MO>. For continuing agents, the input motion token is simply the previously predicted token $\mu_{i,t-1}$.

Each prediction head operates only on its associated tokens. This modular structure allows SceneStreamer to handle heterogeneous outputs while maintaining unified sequence modeling.

## 2.3 MODEL DETAILS

**Token Group Attention.** We design a token group attention mechanism, ensuring the causality while allowing effective information communication. As shown in Fig. 2(B), the rules are (1) tokens within the same group can attend to each other freely (e.g., motion tokens attend to other motion tokens at the same step); (2) tokens belonging to the same object (agent or traffic light) in a later step can attend to the tokens of the same object earlier; and (3) every group of tokens can attend to the existing contexts at current or last step. For example, <MO> can attend to current <TL>. <TL> can attend to <MO> at last step, etc.

**Relative Attention.** We use relative attention biases between tokens, computed from $(\Delta x, \Delta y, \Delta \psi, \Delta t)$, to modulate attention weights, following previous work on query-centric attention (Zhou et al., 2023; Shi et al., 2023; Wu et al., 2024). This makes the input token sequences unaware of the global temporal-geometric information of the object, which eases model's training. A KNN mask restricts attention to spatial neighbors for scalability.

**Model Architecture.** SceneStreamer adopts an encoder-decoder architecture. The encoder embeds information of all map segments to static map tokens, which are cross-attended by the dynamic tokens in the decoder. The decoder generates heterogeneous output via different prediction heads as we discussed in Sec. 2.2. Each decoder layer combines 1) cross-attention between dynamic tokens and static map tokens with 2) self-attention over the dynamic tokens using a structured group-causal mask Fig. 2(B), enforcing semantic and temporal dependencies across token types. As all prediction heads output categorical distributions, SceneStreamer can be trained end-to-end using cross-entropy loss.

## 3 EXPERIMENTS

We evaluate SceneStreamer on a suite of tasks to assess the quality of its generated scenarios and its utility for downstream applications, particularly reinforcement learning (RL) planner training. Our experiments aim to answer the following questions:

- Does SceneStreamer generate realistic and diverse agent states comparable to real-world logged data?
- Can SceneStreamer serve as a versatile simulation platform for motion prediction and scene generation?
- Does training an RL planner in SceneStreamer-generated scenarios lead to improved performance and robustness compared to log-replay traffic flows?

We conduct experiments on the Waymo Open Motion Dataset (WOMD) (LLC, 2019), a large-scale benchmark for motion forecasting and simulation. WOMD contains scenarios captured at 10Hz, providing 1 second of historical data and 8 seconds of future trajectories per scene. Each scenario includes up to 128 traffic participants (vehicles, cyclists, pedestrians) along with high-definition maps. To reduce computational cost, we downsample each scenario to 2Hz, yielding 19 discrete steps per scene. We use ScenarioNet (Li et al., 2023) to manage data. SceneStreamer is trained to predict *all three types of agents* and all agents in the scenario. Details of hyperparameters, training and testing can be found in the Appendix.

## 3.1 INITIAL STATE QUALITY

To assess the realism of SceneStreamer-generated initial states, we use the Maximum Mean Discrepancy (MMD) metric, a standard measure of distributional divergence in generative model-

Table 1: **Initial state MMD metrics.** [†] These methods have access to future agent trajectories and use them to assist in generating initial states, making them incomparable to our setting, where the model performs state initialization without any future information. [‡] We relax the standard evaluation protocol by computing MMD over all logged agents with arbitrary category (instead of only vehicle agents within 50m of the ego vehicle).

| Method | Position | Heading | Size | Velocity |
|---|---|---|---|---|
| MotionCLIP | 0.1236 | 0.1446 | 0.1234 | 0.1958 |
| TrafficGen | 0.1451 | 0.1325 | 0.0926 | 0.1733 |
| LCTGen | 0.1319 | 0.1418 | 0.1092 | 0.1948 |
| UniGen Joint | 0.1323 | 0.2251 | 0.0831 | 0.1915 |
| UniGen w/ Agent-Centric Road | 0.1217 | 0.1095 | 0.0817 | 0.1679 |
| UniGen w/ Traj. Inputs[†] | 0.1197 | 0.1897 | 0.0826 | 0.1657 |
| UniGen Combined[†] | 0.1208 | 0.1104 | 0.0815 | 0.1591 |
| SceneStreamer w/o AR Decoding | 0.1603 | 0.1646 | 0.1172 | 0.2114 |
| SceneStreamer | 0.1291 | 0.1270 | 0.0743 | 0.1970 |
| SceneStreamer w/o AR Decoding[‡] | 0.3237 | 0.1203 | 0.0630 | 0.1183 |
| SceneStreamer[‡] | 0.2198 | 0.0665 | 0.0279 | 0.0730 |

ing (Mahjourian et al., 2024). Lower MMD indicates closer alignment between generated and real agents. We evaluate under two settings: 1) a *strict* protocol from TrafficGen (Feng et al., 2023), considering only vehicles within 50 m of the ego, and 2) a *relaxed* setting that includes all agents of any type (vehicle, cyclist, pedestrian), offering a more comprehensive view of realism across full-scene.

We compare SceneStreamer to several recent scenario generation methods: (1) *TrafficGen* (Feng et al., 2023): a two-stage framework generating initial states then predicting motions. (2) *LCT-Gen* (Tan et al.): a language-conditioned scenario generator trained on natural language captions. We use the non-conditioned variant of LCTGen as in the UniGen paper. (3) *MotionCLIP* (Tevet et al., 2022): a diffusion-based trajectory generator guided by CLIP-style embeddings, implemented in LCTGen paper. (4) *UniGen* (Mahjourian et al., 2024): a joint model for initial state and trajectory generation using diffusion. (5) *SceneStreamer w/o AR decoding:* To evaluate the importance of SceneStreamer's autoregressive agent state decoding, we implement a simplified ablation where all agent attributes are predicted independently in parallel using separate MLP heads. Each attribute is treated as a categorical variable with its own discrete space and no conditioning is performed between attributes. This resembles flat decoding strategies used in prior work (Feng et al., 2023; Tan et al.), and removes the structured token sequencing that enables causally consistent agent state generation in SceneStreamer.

Table 1 compares SceneStreamer with recent baselines across position, heading, size, and velocity distributions. SceneStreamer achieves competitive performance, especially when using autoregressive (AR) decoding, under the strict evaluation protocol (vehicles only and within 50m). Under the relaxed evaluation setting ([‡]), SceneStreamer continues to produce realistic agents beyond vehicles, demonstrating generalization to pedestrians and cyclists. Note that trajectory-informed baselines are not directly comparable. Methods marked [†] use future information to refine initial states, giving them an unfair advantage over our fully predictive model. We find that SceneStreamer's performance drops notably when AR decoding is disabled, showing the importance of ordered token generation. Without this ordered structure, flat decoding often produces invalid combinations, e.g., a pedestrian on a highway lane or a vehicle with inconsistent orientation and lateral velocity. Our sequential decoding mirrors the causal structure of how agents are realistically introduced into traffic scenes, improving robustness and realism in downstream simulation.

## 3.2 MOTION PREDICTION QUALITY

We next evaluate SceneStreamer as a motion predictor. Given the initial traffic state and agent history, we autoregressively predict future trajectories of all agents over a 8-second horizon. During evaluation, we state-force all agent state tokens and the first two steps of motion tokens (i.e., at $t = 0$ and $t = 0.5$ seconds) and then let the model roll out the remaining steps autoregressively. We compare two versions of SceneStreamer: (1) *SceneStreamer-Motion*: The base version of SceneStreamer that is only trained to predict motion tokens and traffic light tokens; (2) *SceneStreamer-Full*: The

Table 2: **Motion prediction metrics on held-out Waymo validation set.** We evaluate SceneStreamer using standard forecasting metrics for all agents and the designated object of interest (OOI).

| Model | All Agents | | | | | | OOI Agents | | | | | |
|---|---|---|---|---|---|---|---|---|---|---|---|---|
| | ADD ↑ | FDD ↑ | $ADE_{avg}$ ↓ | $ADE_{min}$ ↓ | $FDE_{avg}$ ↓ | $FDE_{min}$ ↓ | ADD | FDD | $ADE_{avg}$ | $ADE_{min}$ | $FDE_{avg}$ | $FDE_{min}$ |
| SceneStreamer-Motion | 2.2115 | 0.2459 | 1.2100 | 0.8730 | 3.5336 | 2.4129 | 5.8517 | 0.5521 | 3.3084 | 2.1905 | 9.7568 | 6.0963 |
| SceneStreamer-Full | 2.6486 | 0.2567 | 1.3382 | 0.9339 | 3.8740 | 2.5379 | 6.9229 | 0.5773 | 3.5842 | 2.3008 | 10.4477 | 6.3302 |

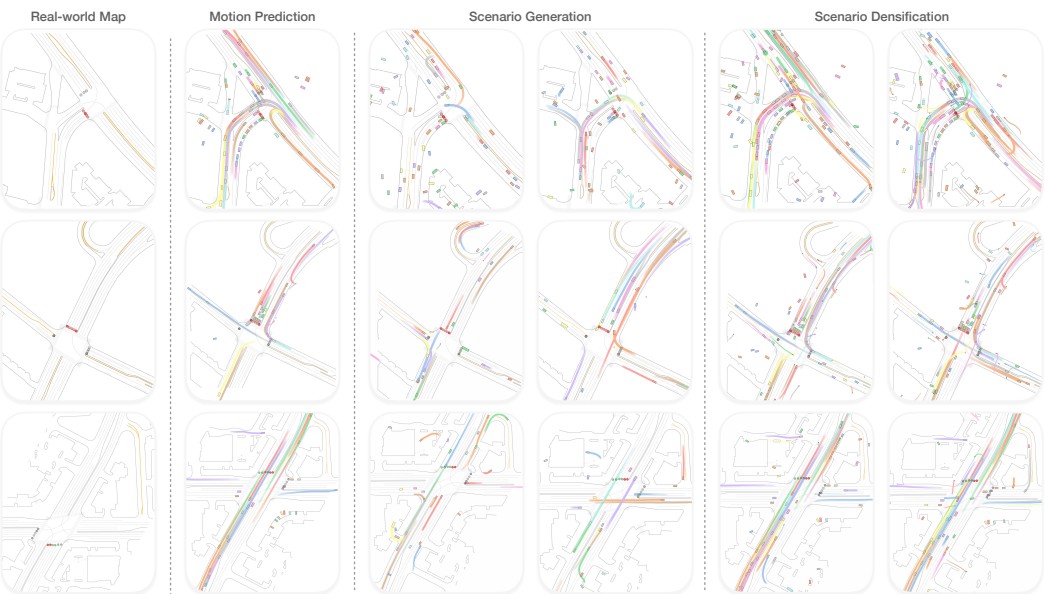

Figure 4: Qualitative results of SceneStreamer in different tasks.

finetuned version of SceneStreamer-Motion that is tasked to predict all dynamic tokens. We evaluate performance on the Waymo validation set using six standard metrics: Average Displacement Error (ADE), Final Displacement Error (FDE), Average Displacement Diversity (ADD), and Final Displacement Diversity (FDD), reported for both all agents and the designated Object of Interest (OOI) defined by the WOMD.

As shown in Table 2, SceneStreamer achieves reasonable motion prediction performance. SceneStreamer-Motion provides accurate predictions with lower ADE/FDE, while SceneStreamer-Full performs slightly worse in accuracy. We hypothesize this is because some attentions from the motion tokens need to be paid to the agent state tokens. Also the capability of the model might be limited due to small parameter size. However, SceneStreamer-Full demonstrates higher ADD and FDD, indicating greater diversity.

### 3.3 QUALITATIVE VISUALIZATION

Fig. 4 illustrates scenes generated by SceneStreamer in different settings, including motion prediction, full-scenario generation, and scenario densification. In the densification task, we state-force the states of existing agents and ask SceneStreamer to generate new agents until 128 agents are reached. We observe that generated agents are well-aligned with map lanes, exhibit coherent motion patterns, and maintain diversity over long horizons. More qualitative visualizations can be found in the Appendix.

### 3.4 PLANNER LEARNING WITH SCENESTREAMER

To evaluate SceneStreamer in downstream autonomous driving (AD), we train reinforcement learning (RL) agents to control the self-driving car (SDC) in SceneStreamer-modified scenarios and test them on unaltered log-replay scenes. This setup examines whether SceneStreamer can serve as a generative simulator that improves planner robustness through diverse, reactive traffic.

Table 3: RL policy performance trained with different traffic simulation sources.

| Training Source | Reward↑ | Success ↑ | Completion ↑ | Off-Road ↓ | Collision ↓ | Cost ↓ |
|---|---|---|---|---|---|---|
| Log-Replay | $32.24_{\pm 3.23}$ | $0.7244_{\pm 0.06}$ | $0.6726_{\pm 0.04}$ | $0.2872_{\pm 0.01}$ | $0.0308_{\pm 0.02}$ | $0.2852_{\pm 0.07}$ |
| SceneStreamer-Motion (No Ada) | $37.98_{\pm 2.50}$ | $0.7355_{\pm 0.05}$ | $0.6783_{\pm 0.04}$ | $0.2940_{\pm 0.02}$ | $0.0270_{\pm 0.01}$ | $0.2795_{\pm 0.02}$ |
| SceneStreamer-Motion (w/ Ada) | $\mathbf{39.23}_{\pm 2.54}$ | $0.7475_{\pm 0.04}$ | $0.7032_{\pm 0.03}$ | $0.2987_{\pm 0.02}$ | $\mathbf{0.0187}_{\pm 0.01}$ | $0.2637_{\pm 0.04}$ |
| SceneStreamer-Full (No RS, No Ada) | $38.18_{\pm 3.01}$ | $0.7339_{\pm 0.05}$ | $0.7052_{\pm 0.03}$ | $0.2932_{\pm 0.03}$ | $0.0194_{\pm 0.01}$ | $0.2697_{\pm 0.04}$ |
| SceneStreamer-Full (No RS, w/ Ada) | $38.81_{\pm 2.30}$ | $0.7385_{\pm 0.05}$ | $0.7230_{\pm 0.01}$ | $0.3010_{\pm 0.03}$ | $0.0290_{\pm 0.03}$ | $0.2880_{\pm 0.07}$ |
| SceneStreamer-Full (w/ RS, w/ Ada) | $39.07_{\pm 2.46}$ | $\mathbf{0.7620}_{\pm 0.04}$ | $\mathbf{0.7345}_{\pm 0.02}$ | $\mathbf{0.2830}_{\pm 0.02}$ | $0.0260_{\pm 0.01}$ | $\mathbf{0.2610}_{\pm 0.03}$ |

Table 4: Waymo Sim Agents Challenge (WOSAC) results on the 2025 test set leaderboard. For all metrics except minADE, higher is better.

| Model | Realism | LinSpd | LinAcc | AngSpd | AngAcc | DistObj | CollLik | TTC | DistEdge | Offroad | minADE |
|---|---|---|---|---|---|---|---|---|---|---|---|
| UniMM (Lin et al., 2025) | 0.7829 | 0.3836 | 0.4160 | 0.5168 | 0.6491 | 0.3910 | 0.9680 | 0.8293 | 0.6791 | 0.9505 | 1.2949 |
| CAT-K (Zhang et al., 2025b) | 0.7846 | 0.3868 | 0.4066 | 0.5203 | 0.6588 | 0.3922 | 0.9702 | 0.8302 | 0.6814 | 0.9524 | 1.3065 |
| SceneStreamer | 0.7731 | 0.3778 | 0.4030 | 0.4232 | 0.5930 | 0.3873 | 0.9694 | 0.8272 | 0.6730 | 0.9467 | 1.4252 |

We use 500 WOMD training scenarios, replacing background traffic with SceneStreamer-generated agents while keeping the SDC trajectory fixed (for computing reward and route completion). Training runs in MetaDrive (Li et al., 2022), which imports scenarios from ScenarioNet (Li et al., 2023) via a unified scenario description format. We convert SceneStreamer outputs into this format, hence seamlessly integrating SceneStreamer with the RL training pipeline. Policies are trained with TD3 (Fujimoto et al., 2018) for 2M steps and evaluated on 100 held-out WOMD validation scenarios. We report standard RL metrics: 1) *Average Episodic Reward*, 2) *Episode Success Rate*: Fraction of episodes that terminate successfully (i.e., reaching goal without major violation), 3) *Route Completion Rate*: Fraction of the predefined route (from GT SDC trajectory) completed per episode, 4) *Off-Road Rate*: Fraction of episodes in which the agent deviates off-road, 5) *Collision Rate*: Fraction of the episodes that have collisions, 6) *Average Cost*: The average number of collisions happen in one episode. Full RL environment details are provided in the Appendix.

We compare several regimes. 1) *Log-Replay* is the baseline trained with unmodified real-world traffic agents. 2) *SceneStreamer-Motion* uses the original initial states of background agents and generates motions of background agents with SceneStreamer. In contrast, 3) *SceneStreamer-Full* generates the initial layout of all agents except SDC, rolls out the motions of background agents, and keeps adding new agents if existing agents leave scene. The variant "adaptive" (*w/ Ada*) means we state-force SDC's trajectory using the latest RL planner's own rollout, otherwise (*No Ada*) SDC follows the ground-truth trajectory. The adaptive version enables the **closed-loop training** for the RL planner (Zhang et al., 2023a): the behavior of SceneStreamer will be influenced by current SDC planner and thus the generated scenarios are conditioned on current RL agent. For SceneStreamer-Full, Reject Sampling (RS) means we regenerate an agent if it collides with existing agents.

Table 3 shows that SceneStreamer-generated scenarios consistently improve planner performance across all metrics. Even without full scenario generation, motion-only variants outperform the log-replay baseline. Adaptive training—where the SDC follows the planner's rollout—further improves robustness and reward. The best-performing setup uses full scenario generation with reject sampling, achieving the highest route completion and lowest cost, demonstrating SceneStreamer's utility as a high-fidelity simulation platform for RL policy training.

### 3.5 WOSAC RESULTS

Table 4 reports performance on the 2025 Waymo Sim Agents Challenge test set. SceneStreamer achieves competitive realism and behavioral likelihood metrics compared to strong baselines such as UniMM (Lin et al., 2025) and CAT-K (Zhang et al., 2025b), which benefit from mixture-of-experts modeling and closed-loop fine-tuning, respectively. Although SceneStreamer does not outperform on minADE, it maintains strong performance across most realism metrics, validating its efficacy as a general-purpose simulator.

## 4 RELATED WORK

**Motion Prediction and Simulation Agents.** Motion prediction models aim to forecast future trajectories of traffic participants given their initial states, maps, and signals. Classical approaches model

agents independently (Chai et al., 2019; Shi et al., 2023) or with joint interaction modeling (Luo et al., 2023; Wang et al., 2023). More recent transformer-based models learn to autoregressively predict motions in an open-loop or semi-closed-loop fashion (Kamenev et al., 2022; Seff et al., 2023; Zhang et al., 2023b; Philion et al.; Hu et al., 2024; Zhou et al., 2024; Zhao et al., 2024). These models typically assume a fixed agent set and focus only on forward rollout, without modifying the initial scene layout. SceneStreamer complements this line of work by modeling both the motion and the generative process of agent state creation, enabling adaptive and evolving agent populations during simulation.

**Scenario Generation.** Scenario generation aims to produce both the initial agent states and their future trajectories. Early methods adopt a two-stage design: generating static snapshots (Feng et al., 2023; Tan et al., 2021; 2023) followed by motion forecasting using a separate module. While effective, such disjoint designs lack shared context across stages and restrict dynamic updates to the agent set. Diffusion-based approaches (Lu et al., 2024; Sun et al., 2024; Chitta et al., 2024; Jiang et al., 2024) learn scene-level generative priors using denoising processes; SceneDiffuser in particular uses a single latent diffusion model for both initialization and rollout with constraint-based control, providing a unified diffusion formulation for driving simulation. Closer to our approach are unified generative simulators that jointly model initial states and motions within one model. Uni-Gen (Mahjourian et al., 2024) jointly generates initial states and motions but performs generation only once at initialization and cannot inject new agents mid-simulation. Concurrent to our work, *Interleaved InfGen* (Yang et al., 2025) uses a single autoregressive model to interleave motion simulation with agent addition, removal, and (re)initialization from a logged 1s seed using ego-centric occupancy grids and a dynamic agent matrix. SceneStreamer instead uses map-anchored token groups shared across initialization, densification, and rollout, enabling scenario generation directly from map tokens (without access to ego state), mid-simulation scene editing, and densification with a single lane-graph-aligned abstraction. Its compact discrete token-group autoregressive transformer with explicit map-anchored state and traffic-light tokens makes it practical as a fast, closed-loop simulator for training reinforcement-learning-based planners. A more comprehensive review of related literature is provided in the Appendix.

## 5 CONCLUSION

We introduced **SceneStreamer**, a unified generative traffic simulator. By representing heterogeneous traffic elements such as vehicles, cyclists, pedestrians, and traffic lights as discrete tokens, SceneStreamer enables flexible, step-by-step simulation of complex traffic scenes. This design enables a wide range of use cases in inference, including motion prediction, scenario densification, and synthetic scene generation, without modifying the model. Unlike prior methods that rely on fixed initial conditions or log-replay agents, SceneStreamer supports dynamic agent injection and closed-loop rollout, facilitating long-horizon and reactive simulations. Shown with extensive experiments, SceneStreamer generates high-fidelity initial states, maintains coherent and diverse traffic behaviors over time, and improves the robustness and generalization of downstream RL planners trained in its generated scenarios.

**Limitations.** SceneStreamer relies on long token sequences to represent dense multi-agent traffic scenes. This leads to high memory demand during training. Another challenge lies in compounding errors during test-time generation. We can opt for recent advances in closed-loop fine-tuning a behavior model to address this issue (Zhang et al., 2025b; Peng et al., 2024; Chen et al., 2025).

ACKNOWLEDGMENTS

This project was supported by the NSF Grants CNS-2235012 and CCF-2344955.

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

# Appendix

APPENDIX CONTENTS

## A  BROADER IMPACT AND SOCIETAL CONSIDERATIONS

SceneStreamer is a generative simulation framework for modeling dynamic traffic scenarios using autoregressive token group prediction. By enabling realistic, reactive, and scalable traffic simulation, SceneStreamer has the potential to significantly advance the development and validation of autonomous driving (AD) systems. This includes improving the robustness of motion planning policies as we studied in the experiment section, facilitating rare event training, and supporting data augmentation in reinforcement learning pipelines.

**Positive Societal Impacts.**   SceneStreamer's ability to generate diverse and reactive traffic scenes can accelerate the safe deployment of AD systems. More robust planners may reduce traffic accidents, improve traffic efficiency, and enhance accessibility for populations with limited mobility. Furthermore, open-sourcing our model and implementation encourages broader research into safety-critical domains without requiring access to expensive real-world data collection or proprietary platforms.

**Potential Negative Impacts and Misuse.**   As a scenario generation tool, SceneStreamer could be misused to simulate rare or malicious driving scenarios for purposes such as adversarial testing without disclosure or crafting unfair benchmarks. Additionally, if used to train agents without proper safety constraints, generated scenarios might lead to overfitting to synthetic patterns or unsafe generalization in deployment. There is also a potential for use in generating deceptive traffic scenes in virtual testing or regulatory submissions.

**Mitigations.**   We emphasize that SceneStreamer is not a closed-loop SDC driving policy and does not dictate real-world behavior. However, we encourage the community to adopt responsible use practices. This includes transparent reporting of synthetic data usage, gating scenario difficulty and validity when used in SDC planner training and evaluation, and coupling SceneStreamer with validation on real-world data. Our open-source release will include documentation clarifying its intended research uses and limitations.

## B EXTENDED RELATED WORK

### B.1 MOTION PREDICTION AND SIMULATION AGENTS

Motion prediction models aim to forecast future trajectories of traffic participants given their past states, road maps, and traffic signals. Classical approaches often treat agents independently (Shi et al., 2022; Chai et al., 2019; Shi et al., 2023; Wang et al., 2024), while more recent models incorporate joint interaction modeling (Luo et al., 2023; Wang et al., 2023; Ding et al., 2024; Suo et al., 2021; Zhang et al., 2022; 2023b; 2024). Transformer-based models have further advanced this field by learning to autoregressively predict motions in open-loop or semi-closed-loop setups (Kamenev et al., 2022; Seff et al., 2023; Philion et al.; Hu et al., 2024; Zhou et al., 2024; Zhao et al., 2024; Lin et al., 2025; Zhang et al., 2025b). In parallel, diffusion models have been introduced as an alternative generative paradigm, including MotionDiffuser (Jiang et al., 2023) and SceneDM (Guo et al., 2023; Chang et al., 2024). Despite impressive progress, these methods operate under the assumption that agents set is fixed and focus solely on the trajectory rollout. They do not modify or expand the initial scene configuration, limiting their use in dynamic simulation. SceneStreamer addresses this limitation by jointly modeling both agent state generation and motion rollout, supporting dynamic agent populations during long-horizon simulation. Furthermore, because they require full access to past and current states for all agents, these motion models are not directly applicable to the scenario generation setting.

### B.2 SCENARIO GENERATION

Scenario generation involves synthesizing both initial conditions and future evolutions of traffic scenes. Procedural generation approaches (Li et al., 2022; Lopez et al., 2018; Dosovitskiy et al., 2017; Zhou et al., 2020; Leurent, 2018; Brunnbauer et al., 2024) rely on hand-coded rules or templates, which limits realism and diversity. Many learning-based works adopt a two-stage pipeline: static scene generation followed by motion forecasting (Feng et al., 2023; Tan et al., 2021; 2023; Cao et al., 2024; Pronovost et al., 2023; Bergamini et al., 2021). For example, SceneGen (Tan et al., 2021) and TrafficGen (Feng et al., 2023) autoregressively add agents based on map anchors, followed by state refinement. SceneStreamer builds on this idea but unifies state and trajectory generation into a single model, promoting global consistency and flexible editing. Diffusion-based methods have also been proposed for full-scene generation (Lu et al., 2024; Sun et al., 2024; Chitta et al., 2024; Rowe et al., 2025), including CTG (Zhong et al., 2023b;a) and SceneDiffuser (Jiang et al., 2024; Tan et al., 2025) which generate dense scenes under language guidance. However, some of these models still separate static and dynamic phases, or generate entire scenes in a single forward pass, limiting interactivity. SceneDiffuser in particular uses a single latent diffusion model for both initialization and rollout with constraint-based control, providing a unified diffusion formulation for driving simulation.

UniGen (Mahjourian et al., 2024) improves on prior work by jointly generating initial states and motion trajectories. However, it generates scenes in a fixed order. UniGen first initializes agent A's state, then predicts the full future trajectory of agent A. Then it initializes the agent B's state, while agent A's future trajectory is accessible. This breaks temporal causality and creates difficulty if we want to conduct closed-loop simulation with it. Moreover, its agent-centric representation requires expensive replanning to maintain closed-loop consistency. In contrast, SceneStreamer generates agent states and motions in a unified token sequence using a single autoregressive model. This enables realistic, causal interactions and allows agents to enter or leave the scene dynamically.

Concurrent to our work, Yang et al. propose InfGen (Yang et al., 2025), which uses a single autoregressive model to interleave motion simulation with agent addition, removal, and (re)initialization. It discretizes agent poses on an ego-centric occupancy grid and expands a dynamic agent matrix with control tokens starting from a logged 1s seed. SceneStreamer differs from these works in several ways. First, it uses map-anchored token groups shared across initialization, densification, and rollout, rather than ego-centric occupancy grids and pose tokens. This provides a single, lane-graph-aligned abstraction for both initial states and motions and supports scenario generation directly from map tokens, as well as mid-simulation scene editing and densification by injecting new agents while keeping the same representation. Second, SceneStreamer is designed as a compact, discrete token-group autoregressive transformer with explicit map-anchored state and traffic-light tokens, enabling direct token-level editing via state-forcing and resampling. This lightweight design makes it practi-

cal to use as a fast, closed-loop simulator for training reinforcement-learning-based planners, where we show that training on SceneStreamer-generated scenarios improves robustness and generalization over log-replay baselines.

### B.3 DATA-DRIVEN SIMULATION

Data-driven simulation environments such as Nocturne (Vinitsky et al., 2022), Waymax (Gulino et al., 2023), MetaDrive (Li et al., 2022), ScenarioNet (Li et al., 2023), and GPUDrive (Kazemkhani et al., 2024) enable scalable simulation by replaying real-world logs. While preserving behavioral realism, the traffic flows in these environments are non-reactive: deviations from the logged trajectory, e.g., when the ego vehicle brakes earlier, can result in implausible interactions like rear-end collisions. Recent advances integrate generative models to create reactive and closed-loop simulation environments. Vista (Gao et al.) predicts future high-resolution images and supports interactive control, while DriveArena (Yang et al., 2024) combines a neural renderer with a physics-based simulator, forming a tight perception-action loop. UniScene (Li et al., 2025) introduces a unified occupancy-centric framework that first generates semantic occupancy from BEV layouts and then conditions on it to synthesize multi-view video and LiDAR, enabling versatile and high-fidelity scene generation. These approaches focus on photorealistic sensor simulation, helping bridge the sim-to-real gap for perception modules.

In terms of interaction-level simulation, works like STRIVE (Rempe et al., 2022) and CAT (Zhang et al., 2023a) generate safety-critical scenarios for safety validation. MixSim (Suo et al., 2023) uses a goal-conditioned policy and actively resimulate different possible goals to enable closed-loop simulation, but its computational cost scales poorly with the number of agents, limiting real-time use. CtRL-Sim (Rowe et al., 2024) applies offline RL to train reactive agents for use in Nocturne, enabling goal-directed, controllable traffic behavior. SceneStreamer complements these works by acting as a fast, flexible scenario generation model that not only generates trajectory but also initializes new agents.

## C   MODEL ARCHITECTURE DETAILS

This section presents the details of **SceneStreamer**: a unified transformer framework that jointly generates traffic-light states, agent initial states, and agent motions in a *single autoregressive token sequence*.

**Our Insights.**   We cast scenario generation as a next-token prediction task: map tokens $\langle \texttt{<MAP>} \rangle$ are followed *each step* by traffic-light tokens $\texttt{<TL>}$, agent-state tokens $\texttt{<AS>}$, and agent-motion tokens $\texttt{<MO>}$ and form a **single autoregressive token sequence**:

$$\mathbf{x}_{1:T} = \big[ \texttt{<MAP>}; (\texttt{<TL>}, \texttt{<AS>}, \texttt{<MO>})_1; (\texttt{<TL>}, \texttt{<AS>}, \texttt{<MO>})_2; \ldots \big].$$

Given all tokens $\mathbf{x}_{<t}$ generated so far, the model predicts the categorical distribution of the next token $p_\theta(x_t \mid \mathbf{x}_{<t})$ and samples $x_t$. Following this idea, we develop **SceneStreamer** learning one transformer that sees the whole history, enabling fine-grained, closed-loop generation and smoother downstream RL integration.

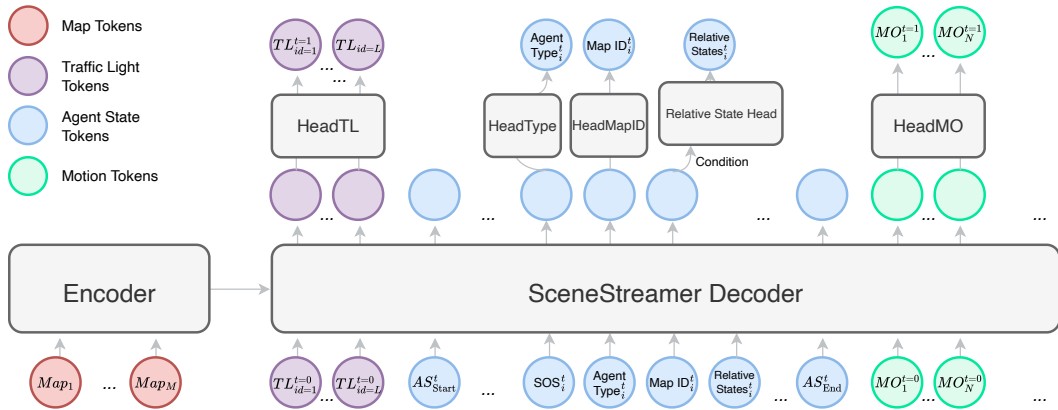

Figure 5: SceneStreamer model architecture.

## C.1 ENCODER-DECODER STRUCTURE

SceneStreamer adopts the encoder-decoder architecture. A lightweight encoder embeds up to 3000 map-segment tokens (produced by slicing each lane-centerline into $\leq$ 10 m segments) into a set of key/value vectors $\mathbf{H}^{\text{map}}$. The output map tokens are later used for cross attention. A decoder autoregressively generates all non-map tokens. In every layer of the decoder, we first conduct self-attention within the input token sequence, where a group attention causal mask illustrated in Fig. 6 is applied. Then we conduct cross-attention between the dynamic tokens and the map tokens.

**Prediction Heads.** As shown in Fig. 5, SceneStreamer uses a shared decoder trunk followed by distinct output heads for each token group. Each head projects the decoder hidden state to a task-specific vocabulary or output space.

(1) *Traffic light head* is an MLP layer HeadTL($\{<\texttt{TL}>_{i,t}\}_{i=1}^{N_{\text{TL}}}$) $\in \mathbb{R}^{N_{\text{TL}} \times 4}$ maps the decoder output to one of four discrete states: $\{\texttt{green}, \texttt{yellow}, \texttt{red}, \texttt{unknown}\}$.

(2) *Agent State Head* is a nested module with several sub-heads and an agent state transformer. We will discuss this in appendix D. Overall, in the agent state generation, two MLPs (the agent type prediction head HeadType and the map ID predictor HeadMapID) as well as a tiny transformer, the Relative State Head, are involved.

(3) *Motion Head:* The motion head predicts each agent's control input as a single categorical token from a 2D discretized space of acceleration and yaw rate. Specifically, we define a flat vocabulary, where each token corresponds to a unique pair $(a, \omega)$ drawn from uniformly quantized grids $\mathcal{A}$ and $\Omega$. We apply a single linear classifier: HeadMO($<\texttt{MO}>_{i,t}$) $\in \mathbb{R}^{1,089}$, followed by a softmax layer to obtain the probability distribution over control tokens. At inference time, we decode the token index using nucleus sampling and map it back to the corresponding $(a, \omega)$ pair via a deterministic lookup table. The predicted control input is then passed through the kinematic update rule to compute the agent's new state.

Each prediction head operates only on its associated tokens, enabled by a token-type embedding and mask within the decoder. This modular structure allows SceneStreamer to handle heterogeneous outputs while maintaining unified sequence modeling.

## C.2 TOKEN EMBEDDINGS AND TYPES

**Map Tokens <MAP>.** A map segment is represented as a polyline, consisting of $N_p$ ordered 2D points with semantic attributes. We denote the set of $M$ map segments in the scene as a tensor $\mathbf{S}_{\text{map}} \in \mathbb{R}^{M \times n \times C}$, where $C$ is the per-point feature dimension (e.g., 2D position, road type one-hot). We adopt the PointNet-like (Qi et al., 2017) polyline encoder yielding map segment features $\{\mathbf{p}_i = \text{PolyEnc}(\mathbf{S}_{\text{map}}^{(i)})\}_{i=1}^{M}$. These are then passed into the SceneStreamer encoder with full self-

attention across map segments:

$$\mathbf{H}^{\text{map}} = \text{SceneStreamer}_{\text{enc}}([\mathbf{p}_1; \ldots; \mathbf{p}_M]) \in \mathbb{R}^{M \times d}. \tag{7}$$

To support cross-attention in the decoder, we assign each map segment a unique discrete index $i$ (its MapID), and embed it into the map token.

$$<\text{MAP}>_i = \mathbf{H}^{\text{map}}[i] + \text{EmbMapID}(i) \otimes \mathbf{g}_i, i = 1, \ldots, M. \tag{8}$$

where EmbMapID is a learned embedding table. $\otimes$ denotes we will record the geometric information $\mathbf{g}_i$ of map segment $i$, which includes its center position and heading, and use it to participate in the relative attention. We defer the discussion of relative attention to appendix C.3.

These enriched map tokens $\{<\text{MAP}>_i\}$ are kept fixed during simulation and serve as static cross-attention keys/values for all decoder layers. Each dynamic token (e.g., $<\text{TL}>$, $<\text{AS}>$, $<\text{MO}>$) performs cross-attention to the map encoder output to incorporate geometric context.

**Traffic Light Tokens $<$TL$>$.** Each traffic light is represented by a single token per step. We encode the traffic light's discrete state (green, yellow, red, or unknown), its unique identifier, and the map segment it resides in $\lambda_k$. Formally, the traffic light token for light $k$ at step $t$ is constructed as:

$$<\text{TL}>_{k,t} = \text{EmbState}(s_{k,t}) + \text{EmbTLID}(k) + \text{EmbMapID}(\lambda_k) \otimes \mathbf{g}_k, k = 1, \ldots, N_{\text{TL}}, \tag{9}$$

where $s_{k,t} \in \{\text{G}, \text{Y}, \text{R}, \text{U}\}$ is the signal state, $\lambda_k$ is the discrete map segment ID the light is attached to, and $\mathbf{g}_k$ is its temporal-geometric context (position, orientation and current timestep). As with map tokens, $\otimes$ indicates that $\mathbf{g}_k$ participates in relative attention (see appendix C.3). All traffic light tokens are generated in a single batch at each step, with the output obtained via a 4-way classification head.

**Agent State Tokens $<$AS$>$.** For every active agent—including newly injected agents at step $t$—SceneStreamer generates a set of four agent state tokens that collectively encode the agent's dynamic and semantic state. These states include positions, headings, velocities, shapes and agent categories. We defer the detailed tokenization and inference process of agent state tokens to appendix D.

**Motion Tokens $<$MO$>$.** To model agent motion, SceneStreamer predicts a tokenized instantaneous control input for each agent, parameterized as a pair of acceleration and yaw rate: $(a, \omega) \in \mathcal{A} \times \Omega$ (Zhao et al., 2024), where $\mathcal{A}$ and $\Omega$ are discretized into 33 uniform bins respectively, covering acceleration and yaw rate ranges observed in training data. This results in a total of $33 \times 33 = 1,089$ motion classes. Given an agent's current state at time $t$: position $(x_t, y_t)$, heading $\psi_t$, and speed $v_t$, the next-step state is predicted using a first-order bicycle-model update over a small timestep $\Delta t$ (we use $\Delta t = 0.5$s):

$$\psi_{t+1} = \psi_t + \omega \cdot \Delta t, \tag{10}$$

$$v_{t+1} = v_t + a \cdot \Delta t, \tag{11}$$

$$x_{t+1} = x_t + v_{t+1} \cdot \cos(\psi_{t+1}) \cdot \Delta t, \tag{12}$$

$$y_{t+1} = y_t + v_{t+1} \cdot \sin(\psi_{t+1}) \cdot \Delta t. \tag{13}$$

We assume zero lateral slip and no wheelbase constraint (i.e., velocity direction aligns with heading).

To obtain the ground-truth motion token for agent $i$ at step $t$, we enumerate all 1,089 candidate $(a, \omega)$ combinations, apply the above update rule to generate candidate next poses, and evaluate them against the true bounding box at $t + 1$. Specifically: (1) For each candidate motion, compute the predicted pose $(x_{t+1}, y_{t+1}, \psi_{t+1})$. (2) Generate the 4 corners of the agent's oriented bounding box based on its shape and predicted pose. (3) Compute the **Average Corner Error (ACE)** as the mean $\ell_2$ distance between predicted and ground-truth corners. (4) Select the $(a, \omega)$ pair minimizing ACE as the ground-truth label $\mu_i$.

This strategy ensures that both position and heading are tightly aligned during supervision. Compared to velocity- or displacement-based tokenization schemes (Wu et al., 2024; Philion et al.), our control-based formulation provides a smoother interpolation of motion intent and better supports

maneuver modeling such as lane changes and turns. It also enables compact tokenization with high spatial precision.

Each motion token $<$MO$>$ corresponds to one agent at a specific timestep and encodes both its control input and identity-related context. Formally, given an agent $i$ at step $t$, we define its motion token embedding as:

$$<\text{MO}>_{i,t} = \text{EmbMotion}(\mu_i) + \text{EmbType}(c_i) + \text{EmbAID}(i) + \text{EmbVel}(\mathbf{v}_i) + \text{EmbShape}(\mathbf{s}_i) \otimes \mathbf{g}_i \tag{14}$$

where $\mu_i$ is the GT motion label selected from 1,089 candidates, $c_i$ is the categorical for agent type (e.g., vehicle, pedestrian, cyclist), $i$ is agent's ID, $\mathbf{v}_i$ is a 2D vector representing the agent velocity in local frame, and $\mathbf{s}_i$ is a 3D vector of agent's shape (length, width, height). $\mathbf{g}_i$ is a 4D vector encoding temporal-geometric information (agent's current global position, heading and time step). At an agent's first appearance, a special label $\mu_{\text{start}}$ is used to get $<$MO$>$. For continuing agents, the input motion token is simply the token with previously predicted motion label $\mu_{i,t-1}$. This enriched representation ensures that motion tokens carry sufficient context for the decoder to generate informed predictions—capturing both semantic (who the agent is) and physical (how it moves) characteristics. The geometric context $\mathbf{g}_i$ also enables relative attention with map and agent tokens, as discussed in appendix C.3.

## C.3 RELATIVE POSITIONAL ATTENTION

Let $x_i, x_j \in \mathbb{R}^d$ be two input tokens in a Transformer layer, where token $x_i$ attends to token $x_j$. Let their geometric or temporal relation be denoted as $(\Delta x_{ij}, \Delta y_{ij}, \Delta \psi_{ij}, \Delta t_{ij})$, computed from their respective spatial anchors and time indices.

In the attention mechanism, we compute the following projections:

$$q_i = \text{MLP}_Q(x_i), \quad k_j = \text{MLP}_K(x_j), \quad v_j = \text{MLP}_V(x_j), \tag{15}$$

$$q_i' = \text{MLP}_{Q'}(x_i), \quad r_{ij} = \text{MLP}_{\text{rel}}(\Delta x_{ij}, \Delta y_{ij}, \Delta \psi_{ij}, \Delta t_{ij}), \tag{16}$$

where $q_i/q_i', k_j, v_j \in \mathbb{R}^{d_h}$ are standard content-based query, key, and value vectors, while $r_{ij} \in \mathbb{R}^{d_h}$ encodes the relation-aware components.

The final attention score is computed as:

$$\alpha_{ij} = \frac{1}{\sqrt{d}} \left( q_i^\top k_j + q_i'^\top r_{ij} \right) + m_{ij}, \tag{17}$$

where $m_{ij} \in \{-\infty, 0\}$ is an attention mask determined by causal constraints and group-level attention rules (see Fig. 6). This formulation introduces spatial-temporal awareness by allowing each query to attend differently depending on its learned relation to the key, improving inductive bias and facilitating structured interactions in traffic scenes (Zhou et al., 2023; Shi et al., 2023; Wu et al., 2024).

**KNN pruning for scalable attention.** To improve scalability in large scenes, we optionally apply K-nearest neighbor (KNN) masking on attention if both query and key tokens carry relative positional information. Specifically, when both tokens are equipped with geometric anchors $\mathbf{g}_i$ and $\mathbf{g}_j$, we compute Euclidean distance in their x-y position and retain only the top-$k$ closest keys for each query. This reduces the attention cost from $O(N^2)$ to $O(Nk)$, while still preserving local interactions that matter for driving behavior. For tokens lacking spatial grounding (e.g., $<$SOS$>$ or $<$TYPE$>$), full attention is retained.

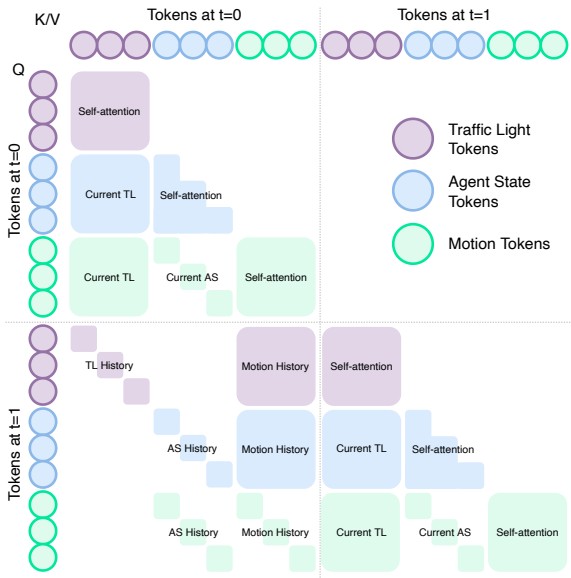

Figure 6: **The attention mechanism in SceneStreamer.** Tokens at each step are grouped into traffic-light (purple), agent-state (blue), and motion (green) tokens. Within a timestep, attention flows from earlier groups to later groups, enforcing semantic causality. Cross-timestep attention allows history tokens to influence current predictions. Empty regions represent masked attention.

### C.4 TOKEN GROUP ATTENTION MECHANISM

As shown in Fig. 6, we enforce structured inter-step attention via a causal group mask: (1) Tokens within the same group can attend to each other freely (e.g., motion tokens attend to other motion tokens at the same step). (2) The tokens belong to the same object (agent or traffic light) in later step can attend to the tokens belonging to the same object earlier. (3) Every group of token can attend to some existing contexts, for example $<$MO$>$ can attend to current $<$TL$>$. Figure 6 illustrates the structured attention mask applied in the decoder. Each quadrant corresponds to a token group at timestep $t = 0$ or $t = 1$ attending to other tokens. The diagonal blocks represent full self-attention within each group, while the off-diagonal regions encode allowed causal flows across groups. For example, at $t = 1$, motion tokens can attend to agent-state and traffic-light tokens from both $t = 0$ and $t = 1$, but not vice versa. This reflects the natural temporal and semantic ordering in generative traffic scenes and helps enforce proper dependency structure during autoregressive decoding.

### D AGENT STATE TOKENIZATION

As shown in Fig. 7(A), each agent $i$ present at step $t$ is represented by four ordered tokens

$$\langle <\text{SOA}>_i, <\text{TYPE}>_i, <\text{MS}>_i, <\text{RS}>_i \rangle_t.$$

Here $<$SOA$>$ is the start-of-agent flag, $<$TYPE$>$ is the categorical token in $\{\text{veh}, \text{cyc}, \text{ped}\}$, $<$MS$>$ is the index of a map segment, $<$RS$>$ is the relative states of agent w.r.t. the selected map segment.

Concretely,

$$<\text{SOA}> = \text{EmbIntra}(4i) + \text{EmbAID}(i) + \text{EmbSOA}, \tag{18}$$

$$<\text{TYPE}> = \text{EmbIntra}(4i + 1) + \text{EmbAID}(i) + \text{EmbType}(c_i), \tag{19}$$

$$<\text{MS}> = \text{EmbIntra}(4i + 2) + \text{EmbAID}(i) + \text{EmbType}(c_i) + \text{EmbMapID}(\lambda_i) \otimes \mathbf{g}(<\text{MAP}>_{\lambda_i}), \tag{20}$$

$$<\text{RS}> = \text{EmbIntra}(4i + 3) + \text{EmbAID}(i) + \text{EmbType}(c_i) + \text{EmbMapID}(\lambda_i) + \text{EmbRS}(\mathbf{r}_i) \otimes \mathbf{g}_i, \tag{21}$$

EmbIntra$(4i + j)$ encodes the intra-step offset of the $j$-th token within agent $i$'s group, EmbAID$(i)$ provides a consistent agent identity embedding reused across steps, EmbType$(c_i)$ represents the

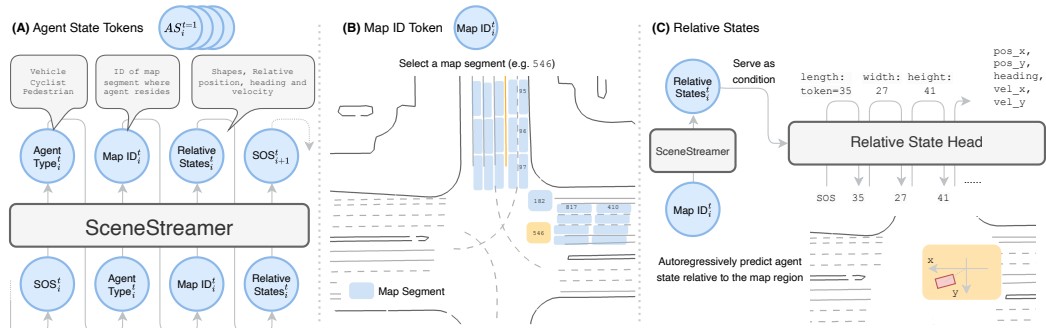

Figure 7: **The design of agent state generation.** **(A)** Each agent's state is encoded as 4 tokens. We first predict the agent type, select a map ID where the agent resides, then predict the relative states. **(B)** Before obtaining the agent state, we first select a map segment as the "anchor" where the agent should reside. **(C)** Feeding in the Map ID, we use the output token as the condition and call the Relative State Head, which is a tiny transformer, to autoregressively generate the relative agent states, including shape, position, heading and velocity.

agent's semantic class, $\text{EmbMapID}(\lambda_i)$ embeds the discrete map segment index $\lambda_i$, $\text{EmbRS}(\mathbf{r}_i)$ embeds the agent's relative state $\mathbf{r}_i$, including position, heading and velocity offsets with respect to the selected map segment and the agent's shape, $\mathbf{g}(\texttt{<MAP>}_{\lambda_i})$ retrieves the geometric anchor (position, heading and current step) of the selected map segment, which participates in relative attention, $\mathbf{g}_i$ denotes the generated agent's current temporal-geometric information.

As shown in Fig. 7 (B), by reading $\texttt{<TYPE>}$, the model will select one of the map segments $\lambda_i$. This is done by applying a map ID head on the output token and conducting softmax sampling on the output logits:

$$\lambda_i \sim \text{Softmax}(\text{HeadMapID}(\text{SceneStreamerDec}(\texttt{<TYPE>}))). \tag{22}$$

As illustrated in Fig. 7 (C), after selecting a map segment $\lambda_i$ and generating the associated $\texttt{<MS>}$ token, we condition on the decoder output of $\texttt{<MS>}$ to generate the agent's full kinematic and shape attributes. Specifically, a dedicated module called the relative state head, a small Transformer decoder with AdaLN (Perez et al., 2018) normalization, is used to autoregressively generate a sequence of 8 tokens (preceded by a SOS token), each representing a field in the agent's relative state vector:

$$\mathbf{r}_i = (\text{SOS}, l, w, h, u, v, \delta\psi, v_x, v_y), \tag{23}$$

where SOS is the start-of-sequence indicator, $(l, w, h)$ is the agent's physical dimensions (length, width, height), $(u, v)$ is the longitudinal and lateral offset from the centerline of map segment $\lambda_i$, $\delta\psi$ is the heading residual relative to the segment orientation, $(v_x, v_y)$ is the velocity whose direction is in the frame of the map segment. Each field is discretized into 81 uniform bins and modeled as a classification problem.

We should mention that there are two special tokens as shown in Fig. 5, the "agent state generation starts" and "agent state generation ends" token, before and after the agent state generation of all agents.

During training, teacher forcing is used to feed ground-truth relative state tokens, while at inference, we apply softmax sampling to decode each dimension sequentially. Within the relative state head, the input at each decoding step consists of the embedding of last selected action out of the vocabulary, added to a learned positional embedding that encodes its index in the sequence. This structure enables fully autoregressive decoding over the 9-token relative state sequence. Unlike prior methods such as TrafficGen (Feng et al., 2023), which generate all agent state attributes simultaneously in a flat and unstructured output head, SceneStreamer decomposes the generation into an ordered, interpretable sequence. This is critical for ensuring semantic and physical consistency. Specifically: Agent type must be sampled first, as it determines downstream constraints on map segment validity, shape bounds, and behavior priors. Map segment selection follows, as it anchors the agent in the environment and defines the frame for relative offset decoding. Relative position $(u, v)$ is then

generated in the local frame of the selected map segment. Heading and velocity are decoded last, conditioned on the selected geometry and pose to avoid implausible combinations. Without this ordered structure, flat decoding often produces invalid combinations, e.g., a pedestrian on a highway lane or a vehicle with inconsistent orientation and lateral velocity. Our sequential decoding mirrors the causal structure of how agents are realistically introduced into traffic scenes, improving robustness and realism in downstream simulation. This compact, conditioned decoding ensures that agents are initialized in contextually appropriate map segments with semantically valid shapes, poses, and velocities. The output relative state tokens are then concatenated and passed back to the main decoder to form the final <RS> token.

**Real-world conversion.** The tokenized agent state is decoded into a global pose and velocity using the geometry of the selected map segment. Given the segment pose $(x_\lambda, y_\lambda, \psi_\lambda)$ and the predicted relative offset $(u, v, \delta\psi, v_x, v_y)$ in the local frame of the segment, the agent's global state is computed as:

$$x = x_\lambda + u \cos \psi_\lambda - v \sin \psi_\lambda, \tag{24}$$

$$y = y_\lambda + u \sin \psi_\lambda + v \cos \psi_\lambda, \tag{25}$$

$$\psi = \psi_\lambda + \delta\psi, \tag{26}$$

$$v_x^{\text{global}} = v_x \cos \psi_\lambda - v_y \sin \psi_\lambda, \tag{27}$$

$$v_y^{\text{global}} = v_x \sin \psi_\lambda + v_y \cos \psi_\lambda. \tag{28}$$

At inference time, new agent state token groups are generated via autoregressive sampling. If the resulting $(x, y)$ position lies within an occupied region or causes overlap with existing bounding boxes, the sampled <MS> or <RS> tokens are rejected and resampled up to a fixed number of retries. A maximum of $N_{\text{new}}$ agents can be injected per step.

For existing agents that persist from the previous timestep, SceneStreamer bypasses the relative state prediction head and instead deterministically generates their agent state token group using their observed global state. Specifically, we first identify the closest map segment $\lambda_i$. Then we compute the relative state $(u, v, \delta\psi, v_x, v_y)$ by transforming the agent's global pose and velocity into the local frame of $\lambda_i$. These values are used to obtain the corresponding <RS> token. The four agent state tokens (<SOA>, <TYPE>, <MS>, and <RS>) can then be constructed directly via embedding lookup and state-forcing. This allows SceneStreamer to seamlessly unify dynamic agent injection (via sampling) and agent motion continuation (via projection), ensuring closed-loop autoregressive simulation across variable-length agent sets.

## E  TRAINING AND INFERENCE DETAILS

### E.1  DATASET AND PREPROCESSING

**Map Preprocessing.** We preprocess the vectorized HD map into a fixed-length token representation by segmenting the raw polylines into discrete map segments. Each polyline is split into segments of approximately 10 meters in length. To limit memory and computational cost, we cap the total number of segments to 3000 per scene. If the number of segments exceeds 3000, we sort all segments by their Euclidean distance to the SDC's current position and retain the closest 3000 segments.

Each segment consists of up to 30 points and is represented by a 27-dimensional feature vector per point. The segment-level position and heading are computed by averaging the position and heading of all points in the segment. The point-level features include geometric information, heading encoding, and semantic labels derived from MetaDrive map types. Specifically, each point feature contains:

- Start and end coordinates: 6 dimensions $(x_s, y_s, z_s, x_e, y_e, z_e)$
- Direction vector: 3 dimensions $(dx, dy, dz)$
- Heading: raw heading, sine, cosine (3 dimensions)
- Point length (1 dimension)

- Binary map type indicators (12 dimensions): `is_lane`, `is_sidewalk`, `is_road_boundary_line`, `is_road_line`, `is_broken_line`, `is_solid_line`, `is_yellow_line`, `is_white_line`, `is_driveway`, `is_crosswalk`, `is_speed_bump`, `is_stop_sign`
- Segment length (1 dimension)
- Valid mask (1 dimension)

Formally, the full feature vector for each map point is a 27-dimensional vector:

$$\mathbf{f} = [x_s, y_s, z_s, x_e, y_e, z_e, dx, dy, dz, \theta, \sin(\theta), \cos(\theta), l, t_1, \ldots, t_{12}, L, m], \tag{29}$$

where $l$ is the segment length, $t_i$ are binary indicators for map semantics, $L$ is total road length, and $m$ is a binary mask indicating validity. The processed segments are stored as a tensor of shape $[M, 30, 27]$ accompanied by a binary mask of shape $[M, 30]$ for downstream consumption in the Transformer encoder.

**Traffic Light Preprocessing.** We preprocess traffic light tokens from the raw data by extracting their spatial and semantic states over time and aligning them with the map representation. As we discussed in appendix C.2, for each traffic light, we will prepare this information:

- the traffic light ID (index),
- the map segment index it is attached to,
- the traffic light state (semantic),
- the position of its stop point (spatial),
- and its heading aligned with the associated map segment.

The ground truth prediction for a traffic light token at each timestep is its state in the next step, formulated as a 4-way classification problem (unknown, green, yellow, red).

**Agent and Motion Preprocessing.** We only select agents that are valid at $t = 10$, which is designated as the "current" step in Waymo Open Motion Dataset (WOMD).

Agents are reordered based on type so that vehicles appear first, followed by pedestrians and then cyclists.

To improve training efficiency, we introduce a configurable maximum agent count $N$. If a scene contains more than $N$ agents, we rank all agents by their cumulative movement distance and retain the top-$N$ most dynamic ones. The remaining agents are masked out at all timesteps, reducing the number of tokens processed per scene.

For each agent, we extract the following attributes:

- Agent ID (used for a dedicated ID embedding),
- Agent type (vehicle, pedestrian, or cyclist),
- Agent shape (length, width, height) at the current timestep,
- Agent position and heading at each timestep (used to locate tokens spatially),
- A 2D velocity vector in the agent's local frame.
- The motion label in $33 \times 33 + 1 = 1090$ candidates (one of them is $\mu_{\text{start}}$).

This information is sufficient for constructing motion tokens as described in the model architecture. Agent motion labels are generated following the tokenization scheme in appendix C.2. At the first timestep when an agent becomes valid, a special start label $\mu_{\text{start}}$ is used to generate its first motion token `<MO>`. For subsequent steps, the model uses the previous token $\mu_{i,t-1}$ as autoregressive input. The ground truth motion label is defined as the motion label at the next step. We skip loss computation for any motion token if the agent is invalid at the current step or if the next-step label is unavailable due to the agent becoming invalid at the following timestep.

**Agent State Ground Truth Preprocessing.** Agent state tokens are used to autoregressively generate new agents into the scene during scenario generation. These tokens encode where, what, and how to instantiate an agent within the current simulation state.

At each sparse timestep (sampled every 5 steps in WOMD), we iterate over all valid agents and compute token values and features as follows:

- Closest Map ID: For each agent, we identify the nearest valid map segment based on Euclidean distance and relative heading difference. Only map features with angular deviation less than $90°$ are considered valid.
- Relative Feature Encoding: The agent's state is expressed as a 8D vector relative to the closest map segment:
    - 2D position offset rotated into the local map frame,
    - heading difference relative to the segment heading,
    - velocity vector rotated into the map frame,
    - agent shape (length, width, height).
- Agent ID: Each agent is assigned a unique ID from 0 to $N - 1$, where $N$ is the number of agents in the scene.
- Intra-step Index: We assign each token a unique intra-step index in $\{0, \dots, N \times 4 + 1\}$ to support position embeddings for agent state autoregressive generation.
- Agent Type: The semantic category of each agent (e.g., vehicle, pedestrian, cyclist) is included as a discrete token input.

The relative feature of the agents are also discretized into 81 uniform bins and serve as the input as well as the GT for the Relative State Head. These components are later embedded and combined via additive token fusion as described in appendix D to form the final agent state token representation.

### E.2 TOKENIZATION HYPERPARAMETERS

All dynamic tokens in SceneStreamer are represented as discrete entries in their respective vocabularies, akin to words in a language model. Each token type has a dedicated tokenization scheme with different vocabulary sizes and resolution bounds.

**Traffic Light Token.** Traffic light tokens represent the current state of a traffic signal. They are selected from a fixed vocabulary of 4 discrete states:

- `0`: Unknown,
- `1`: Green,
- `2`: Yellow,
- `3`: Red.

**Motion Token.** Motion tokens discretize the space of continuous action commands. Each motion token corresponds to a tuple $(a, \omega)$ where $a$ is acceleration and $\omega$ is yaw rate. Both are quantized into 33 bins linearly spanning their respective ranges:

- Acceleration $a \in [-10, 10]$ m/s$^2$,
- Yaw rate $\omega \in [-\frac{\pi}{2}, \frac{\pi}{2}]$ rad/s.

This results in a vocabulary of $33 \times 33 = 1089$ regular motion tokens, plus one special $\mu_{\text{start}}$ token, yielding a total vocabulary size of 1090.

**Agent State Token.** Agent state tokens are composed of three tokens:

- **Agent Type:** chosen from 3 categories:
    - `0`: Vehicle,
    - `1`: Pedestrian,
    - `2`: Cyclist.
- **Map ID:** selected from up to 3000 valid candidate map segments per scene.
- **Relative State Feature:** an 8-dimensional vector. Each bin is indexed into a shared vocabulary of size 81 per dimension, and all attributes are tokenized independently. The binning bounds are:

$$
\begin{aligned}
\texttt{position\_x}, \texttt{position\_y} \quad &\in [-10, 10] \text{ m} \\
\texttt{velocity\_x} \quad &\in [0, 30] \text{ m/s} \\
\texttt{velocity\_y} \quad &\in [-10, 10] \text{ m/s} \\
\texttt{heading} \quad &\in \left[-\tfrac{\pi}{2}, \tfrac{\pi}{2}\right] \text{ rad} \\
\texttt{length} \quad &\in [0.5, 10] \text{ m} \\
\texttt{width} \quad &\in [0.5, 3] \text{ m} \\
\texttt{height} \quad &\in [0.5, 4] \text{ m}
\end{aligned}
$$

### E.3    MODEL TRAINING AND INFERENCE DETAILS

**Loss Function.**    All prediction heads in SceneStreamer are trained using the standard cross-entropy loss. For motion and agent state tokens, loss is only applied to valid entries. Specifically, we exclude tokens if the agent is invalid at the current timestep or if the corresponding ground truth (e.g., next-step motion) is undefined.

**Training Schedule.**    SceneStreamer is trained in two stages:

- Pretraining: We first train a base model using only traffic light and motion tokens. The agent state decoder is disabled during this phase.
- Finetuning: We then finetune the pretrained model with the agent state decoder enabled, jointly predicting agent state, traffic light, and motion tokens.

Pretraining runs for 30 epochs with a batch size of 4, while finetuning runs for 5 epochs with a batch size of 1. Early stopping is applied in the finetuning phase if the minJADE motion metric begins to degrade. We use the AdamW optimizer with a learning rate of 0.0003, cosine decay schedule, 2000 warmup steps, no weight decay, and gradient clipping with a max norm of 1.0. During training, we allow maximally 36 (pretraining) or 28 (finetuning) agents in a scenario to avoid GPU running out of memory. During inference of course we allow maximally 128 agents.

**Hardware.**    All models are trained on 8 NVIDIA RTX A6000 GPUs, each with 48 GB of memory. Pretraining takes approximately 5 hours per epoch, while finetuning takes about 12 hours per epoch due to the added complexity and reduced batch size.

**Inference.**    At inference time, we sample motion tokens using nucleus sampling with $\texttt{topp} = 0.95$. For all other token types (e.g., traffic light, agent state), we use softmax sampling.

**Model Architecture.**    The encoder consists of 2 layers and the decoder has 4 layers. The model uses a hidden dimension $d_{\text{model}} = 128$ with 4 attention heads. The full model has approximately 4.6 million parameters, while the base model (excluding agent state components) has 3.3 million parameters.

### E.4    REINFORCEMENT LEARNING SETUP

**MetaDrive RL Environment.**    We use MetaDrive (Li et al., 2022) ScenarioEnv, which supports loading scenario descriptions (SD) generated by ScenarioNet (Li et al., 2023). Since SceneStreamer also takes SD as input, it is straightforward to implement a bidirectional converter to integrate SceneStreamer outputs into MetaDrive's simulation environment. To enable closed-loop training, we implement a pipeline that converts predicted agent states from SceneStreamer into ScenarioNet SD format. MetaDrive APIs are then used to set the simulation scenario dynamically. During training, we maintain a buffer that stores the ego agent's past trajectory when it previously encountered the same scenario. This trajectory is embedded into the SD before being passed to SceneStreamer. During SceneStreamer inference, we state-force the ego agent's states and actions, generating a scenario that reflects the most recent policy behavior. The resulting SD is then sent back to MetaDrive for simulation and policy training.

**Task Setting.**    The task is defined as following the trajectory of the self-driving car (SDC) while driving as fast as possible and avoiding collisions. In the SD sending to MetaDrive, we always overwrite the SDC's trajectory by the original trajectory, thus the reward and route completion are always computed against the GT SDC trajectory.

**Observation Space.** The RL agent receives the following observation at each timestep:

1. A 120-dimensional vector representing lidar-like point clouds within a $50\,m$ radius around the agent. Each value lies in $[0, 1]$ and encodes the normalized distance to the nearest obstacle in a specific direction, with added Gaussian noise.

2. A vector summarizing the agent's internal state, including steering, heading, velocity, and deviation from the reference trajectory.

3. Navigation guidance in the form of 10 future waypoints sampled every $2\,m$ along the reference trajectory, transformed into the agent's coordinate frame.

4. A 12-dimensional vector for detecting boundaries of drivable areas (e.g., solid lines, sidewalks) using similar lidar-based point clouds.

The ultimate observation is a 161-dimensional vector. The setting follows the original ScenarioEnv setting (Li et al., 2023).

**Action Space** The policy is an end-to-end controller producing a continuous two-dimensional action vector $a \in [-1, 1]^2$, which is scaled and clipped into throttle/brake force and steering angle commands.

**Reward Function** The total reward is composed of four terms:

$$R = c_1 R_{\text{disp}} + c_2 P_{\text{smooth}} + c_3 P_{\text{collision}} + R_{\text{term}}. \tag{30}$$

- Displacement reward: $R_{\text{disp}} = d_t - d_{t-1}$, where $d_t$ denotes the longitudinal progress along the reference trajectory in Frenet coordinates.
- Smoothness penalty: $P_{\text{smooth}} = \min(0, 1/v_t - |a[0]|)$ penalizes sudden steering at high velocity $v_t$, where $a[0]$ is the steering control.
- Collision penalty: $P_{\text{collision}} = 2$ for collisions with vehicles/humans, and $0.5$ for static objects (e.g., cones, barriers).
- Terminal reward: $R_{\text{term}} = +5$ for successful arrival, $-5$ if the agent ends $> 2.5\,m$ from the reference trajectory.

We set $c_1 = 1$, $c_2 = 0.5$, and $c_3 = 1$ in all experiments.

**Termination Conditions and Evaluation** Episodes terminate under the following conditions:

1. The agent deviates >4m from the reference trajectory (out of road).

2. The agent reaches its destination (success).

3. The agent fails to complete the episode within 100 steps (the Waymo scenario typically has 91 steps).

**Evaluation Metrics:** Policies are evaluated on a held-out validation set of 100 real-world scenarios from the WOMD validation set. We report:

1. Average Episodic Reward: Total accumulated reward.

2. Episode Success Rate: Fraction of episodes that terminate successfully (i.e., reaching goal without major violation).

3. Route Completion Rate: Fraction of the predefined route (from GT SDC trajectory) completed per episode.

4. Off-Road Rate: Fraction of episodes in which the agent deviates off-road.

5. Collision Rate: Fraction of the episodes that have collisions.

6. Average Cost: Combined penalty for collisions and off-road violations.

**RL Training.** We adopt the TD3 algorithm (Fujimoto et al., 2018) implemented in Stable-Baselines3 (Raffin et al., 2021). The training is performed in a continuous control setting using the following hyperparameters:

- `learning_rate`: $1 \times 10^{-4}$

- `learning_starts`: 200 steps
- `batch_size`: 1024
- `tau`: 0.005 (for soft target updates)
- `gamma`: 0.99 (discount factor)
- `train_freq`: 1 (update after every step)
- `gradient_steps`: 1 (one gradient update per environment step)
- `action_noise`: None

## F  FREQUENTLY ASKED QUESTIONS

**Q1: Does the number of agents have to be fixed at every step? How can new agents be inserted?**

The number of agents in SceneStreamer does not need to remain fixed at every step. Agents can be dynamically added or removed by adjusting the corresponding set of agent state tokens, which serve as the representation of the current traffic participants.

At a given time step $t$, the number of agents is determined by how many agent state tokens are present. For example, if there are $N$ agents at step $t$, then one must reconstruct $4N$ Agent State (AS) tokens by constructing the agent states and subsequently tokenizing them. In addition, there will be $N$ motion tokens at this step.

At the next step $t + 1$, new agents can be introduced or existing ones removed by modifying the set of agent state tokens accordingly. Suppose we want to add a new agent at $t + 1$. In this case, after state-forcing the first $4N$ AS tokens corresponding to the existing $N$ agents, SceneStreamer autoregressively generates four additional tokens to construct the initial state of the new agent. Then, during motion generation, the model will produce $(N + 1)$ motion tokens in a single batch—one for each of the existing $N$ agents plus the newly added agent—thereby ensuring that the new agent's motion is seamlessly integrated.

During inference, we maintain an incremental list of agent IDs. When a new agent is introduced, a unique ID is assigned to it. This ID is embedded into the input tokens, acting as a positional embedding that identifies the agent to the model. Because SceneStreamer operates in an autoregressive manner, it is straightforward to accommodate four new tokens for the agent's state and one additional token for its motion. This design allows the model to flexibly expand or shrink the set of agents at any time step without disrupting the overall generation process.

**Q2: How does the model know when to stop inserting new agents?**

Just like the language model which has a `<end_of_sequence>` token to indicate it wants to stop generation, we also have this `<start_of_agent_states>` token and `<end_of_agent_states>` token prior and post to the agent state tokens. The model will produce the `<end_of_agent_states>` after it generates the fourth token of the last agent to indicate the stop of generation. In practice the model does well to produce a reasonable number of agents. In our densification experiment, we manually set the total number of agents to 80 to make as many agents as possible. This can be done by just set the output logit of `<end_of_agent_states>` to -inf. In the scenario generation experiment, following the protocol of Waymo Sim Agent challenge, we will know how many vehicles, pedestrians and cyclists there are in a scenario and we will force the model to generate these agents.

**Q3: What is state-forcing and will this cause information leak in test time?**

In SceneStreamer, "state-forcing" specifically refers to the process of first reconstructing the agent states at current step via the forward kinematics (this can be inferred from the states and the predicted motion at previous step) and then tokenizing the new states into agent state tokens. Then we bypass the agent state generation but instead just append those reconstructed agent state tokens into the input sequence. Thus, this is a completely reasonable setting at inference time, and there is no information leak from ground-truth data. In test time, our model runs autonomously without access to ground-truth data.

# G    QUALITATIVE VISUALIZATIONS

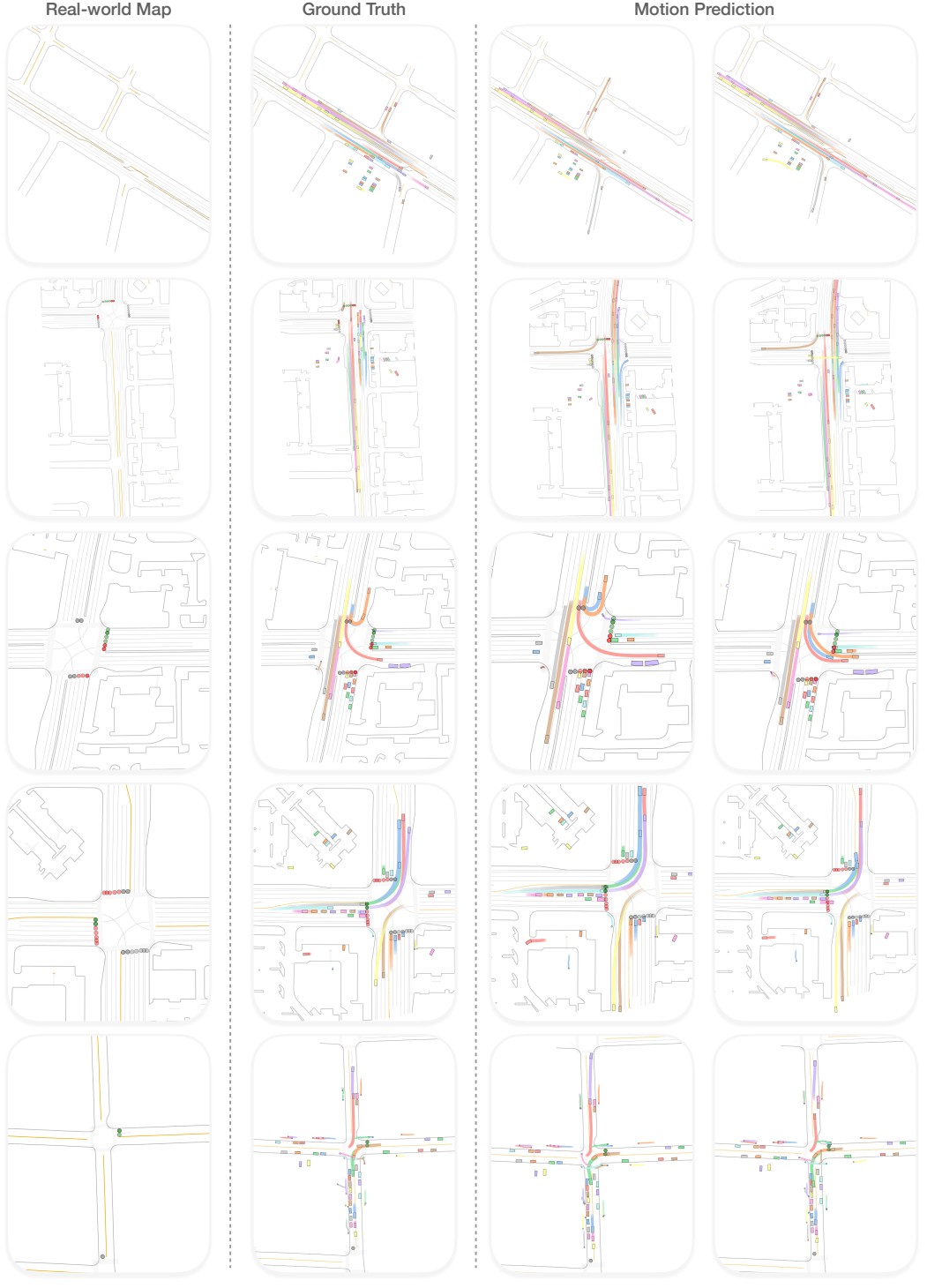

Figure 8: Qualitative visualizations for motion prediction task.

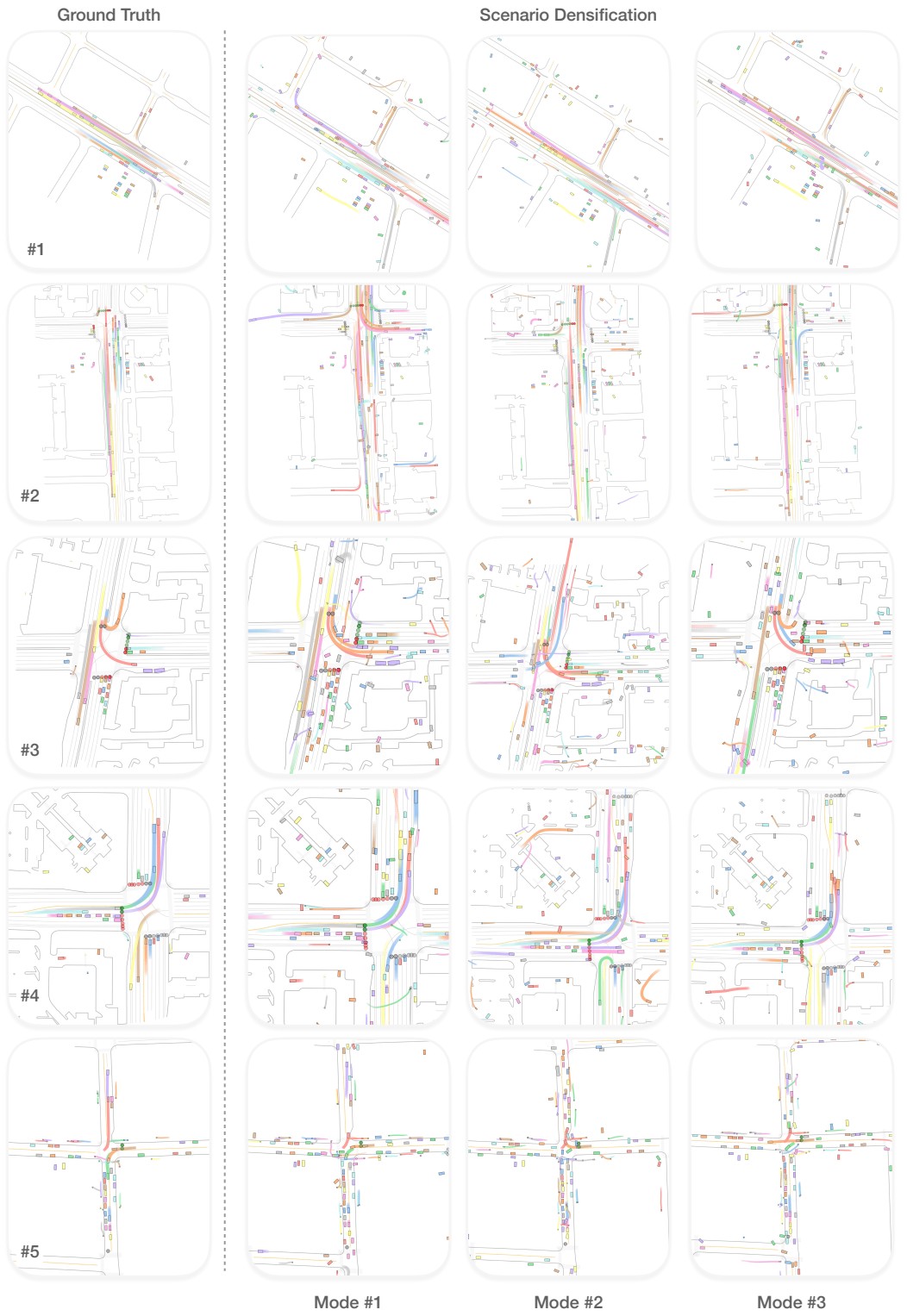

Figure 9: **Qualitative results for scenario densification.** SceneStreamer injects new agents with realistic behaviors such as jaywalking (Scenario #1 Mode #2, S1M2, S3M1), U-turns (S3M3, S4M2), and queueing (S2M2, S3M1). Common failure cases include overspeeding (S2M1), collisions (S4M2, S4M3), and signal violations (S2M1).

