# OpenReview forum: "SceneStreamer: Continuous Scenario Generation as Next Token Group Prediction"
_ICLR.cc/2026/Conference — ICLR 2026 Poster_

### Official Review · Reviewer_h7bB · 2025-10-28

**Soundness:** 3
**Presentation:** 3
**Contribution:** 2
**Rating:** 4
**Confidence:** 5

**Summary:**

This paper presents a framework to employ a single autoregressive model that produces both agent states and their motion trajectories as part of one continuous token sequence over a long horizon. Experiments show the proposed method achieves good performance on Scene Generation and Motion Prediction.

**Strengths:**

1. The unified token scheme for unified motion and scenario generation is interesting (though not super novel).

The proposed method is an interesting way to present the two aspects of traffic simulation, scene generation and motion prediction, in a unified autoregressive architecture. This is an interesting way to scale the full simulation process. However, as will be mentioned in the Weakness section, this idea has been discussed in a prior work.

2. Performance of Scenario Generation and Motion Prediction is good.

The performance of the proposed method on individual tasks of Scenario Generation and Motion Predictions is strong against prior work.

3. Showing improved RL agent performance with InfGen scenarios.

The highlight of this paper's result is its ability to serve as a scenario generator for an effective data augmentation tool for closed-loop simulation.

**Weaknesses:**

1. The main technical contribution is not that novel and misses some important references and discussions.

The main technical contribution of this paper is a unified framework to "employ a single autoregressive model that produces both agents' initial states and their motion trajectories". However, this idea is not new and has already been explored in a prior work already published [1]. Therefore, I think it is necessary to discuss and compare against [1] as it targets the same task and has a very similar idea.

On another level, this paper claimed that prior work like [2] "separates static and dynamic phases", which is not accurate. The method in [2] also achieves both scenario generation and motion prediction with a unified model, during which the model has the full unified context of agent placement and movement information. Although it's a diffusion model instead of an autoregressive model, I also think this paper should discuss its difference with [2], a prior work that unifies the two tasks. Note that [2]  does not do the "fixed-order" generation like UniGen.

2. Does not conduct extensive experiments on many important aspects of the proposed model.

Although the paper shows that the proposed method achieves strong performance for two tasks, there lack of experiments regarding some of the important capabilities mentioned in the main paper:

- Scene editing.
- Agent insertion/removal during simulation.
- Traffic simulation (only in the appendix).
- Extended long-term simulation (no formal experiment, just two demo runs from the supp video).


3. Long-term simulation performance issue from the demo video.

I appreciate the authors presenting the demo video of the scenarios generated by the proposed method. I found the long-term simulation capacity of the proposed method is weak:

- (01:46) The model simulates the ego agent moving forward, but the two agents on the right edge of the scenario cannot exit the scene. These agents drives normally to move out of sight of the ego vehicle, which should be removed by the system. However, the system fails to do that and forces the two vehicles to "come back" to the region of interest, leading to the unsafe collision and off-road behaviors of these two vehicles. I think this might due to the fact that the proposed method does not learn when to remove exiting agents; therefore, to maintain the status of these agents, it forces them to "stay inside" the region of interest. This would become problematic when the simulation duration is longer than the training data.

- Throughout the whole video, I cannot see any agent being added to the scenario. This would be problematic for scenarios like (02:15) where there are less and less agents in the scenario, making the traffic distribution unrealistic.

I wish the author could discuss the reason behind these issues in a systematic way.


4. Minor - Method naming conflict with [1].

The method in [1] is also named InfGen, which has already been published in ICCV. Given that both works focus on traffic or motion generation in simulation environments, this overlap may confuse future citations and discussions. Since the arXiv version of [1] was publicly available months before this submission, the identical naming may lead to confusion when referencing the two works. I suggest the authors consider renaming their method or adding a distinguishing qualifier (e.g., by emphasizing a unique aspect that they can also do initial scenario generation) to avoid ambiguity and ensure clearer attribution.

[1] Long-term Traffic Simulation with Interleaved Autoregressive Motion and Scenario Generation. ICCV 2025.

[2] SceneDiffuser: Efficient and Controllable Driving Simulation Initialization and Rollout. NeruIPS 2024.

**Questions:**

Please refer to the weakness section.

---

> ### Author Response · Authors · 2025-11-21
> **Response to Reviewer h7bB (1/2)**
>
> We thank the reviewer for the detailed and constructive feedback!
>
> ---
>
> ## (W1, W4) Relation to the ICCV paper[1]
>
> We thank the reviewer for pointing us to ICCV paper [1]. After carefully studying [1], we agree that both works share the same high-level idea: using a single next-token autoregressive (NTP) model to jointly handle motion simulation and scenario/agent generation.
> We will clearly acknowledge [1] in the revised version and adjust our claims accordingly.
> In the revision, we will soften and refocus our contribution statement.
>
>
> Conceptually, both works use a single autoregressive model to handle motion and scene-level tokens. However, there are important differences:
>
>
> **1) We use map-anchored, single-vocabulary token groups.**  For each agent we autoregressively decode
> `<TYPE>` → `<MAP-SEGMENT ID>` → `<RELATIVE STATE>` → `<MOTION TOKENS>`
> directly conditioned on a concrete map segment and local kinematics, rather than via pose tokens plus an occupancy grid. This gives a single token-group abstraction for both initial state and motion, tightly tied to the lane graph.
> In contrast, [1] inserts agents via *ego-centric grid-based pose tokens* (discrete position and heading), requires an occupancy grid to decode the new agent w.r.t. the ego car.
>
> **2) Scenario generation from maps and editing.**
> Because everything is expressed as token groups anchored to map segments, we can
> generate full scenarios from map tokens only,
> force / edit subsets of token groups, and
> densify scenes by injecting new agents while keeping the same representation.
> This is different from [1], which always starts from a logged 1s seed and is not used as a “from-scratch” scenario generator. In [1], the new agents are created in the ego-centric occupancy map, so the existance of ego car is assumed.
>
> **3) Light-weight simulator for RL.**
> Our transformer is intentionally small (~4.6 M parameters) and operates on a flat token-group sequence without dynamic agent matrices or occupancy-grid encoders. This makes it easy to plug into RL training as a fast, closed-loop, unified scenario generator; we show that planners trained on our generated rollouts become more robust than log-replay baselines. Note that in [1], **there is no RL experiment demonstrating the benefit of the generated scenarios.**
>
> **4) Task scope is different.**
> [1] is designed specifically for long-horizon WOSAC simulation: given a 1 s log, it focuses on 9–30 s closed-loop rollouts and introduces long-term metrics such as ACE slope and extended Sim Agents scores.
> Our work targets scenario generation and editing on WOMD, including (1) generation from map only, (2) scenario densification, and **(3) closed-loop RL training using our model-generated background traffic, with a compact backbone**. While both works unify scene and motion into one model, our contribution lies in the token-group formulation and map-anchored state generation that make these tasks easy to express (and thus easier to run in closed-loop inference) via simple state-forcing masks.
>
> **5) Traffic light modeling.** Our model also models traffic light states.
>
> We will update the paper to (i) explicitly cite and describe [1], (ii) reorganize the statement of our model and downweight the claim that we are the first unified scenario generation model, and (iii) present our contribution as a different design point: a compact, map-anchored token-group scenario generator that allow streaming of scenario generation.
>
>
> Thank you for flagging the naming collision, and we apologize for the concurrent coincidence. To avoid confusion, in the camera-ready we are happy to rename our method to SceneStreamer and refer to [1] as Interleaved InfGen throughout the paper. We will explicitly state this distinction in the introduction and related work so that future citations can clearly distinguish the two works.

---

> ### Author Response · Authors · 2025-11-21
> **Response to Reviewer h7bB (2/2)**
>
> ## (W1) Relation to SceneDiffuser [2]
>
> You are correct that our current text stating that diffusion-based methods “separate static and dynamic phases” is inaccurate for SceneDiffuser. SceneDiffuser introduces a unified scene-level diffusion prior that handles both scene initialization and rollout via amortized diffusion and constraint-based control.
>
> In the revision we will: 1) Correct the description to acknowledge that SceneDiffuser uses a single latent diffusion model for both initialization and rollout; and
> 2) Clarify the difference: SceneDiffuser operates in a continuous diffusion latent space with amortized denoising and constraint-based control, whereas our model is a discrete token-group autoregressive transformer with explicit map-anchored state tokens and direct token-level editing via state-forcing and resampling.
> 3) We will also remove any implication that SceneDiffuser does not have unified context over initialization and rollout.
>
> ---
>
> ## (W2) Experiments on scene editing, agent insertion/removal, and traffic simulation
>
> We agree that some capabilities are under-emphasized quantitatively.
>
> 1. **Scene editing & agent insertion/removal.** Our model already supports editing via selective state-forcing: for densification, we keep existing state tokens fixed and repeatedly sample additional agent-state tokens until a target count, as illustrated qualitatively in Fig. 4. In the revision we will add quantitative analyses:
>
>   * statistics of inserted agents (positions and headings) versus ground-truth agents on the same map segments;
>   * collision/off-road rates and agent-count distributions for densified scenes versus log replay.
>
> 2. **Traffic simulation in the main paper.** We currently place some WOSAC results in the appendix for space. We will move a concise summary into the main text (Sec. 3.3), reporting collision rate, off-road rate, and route completion when background traffic is fully driven by our model, to better emphasize the simulator role.
>
> 3. **Extended long-term behavior.** Our quantitative evaluation focuses on horizons comparable to WOMD/WOSAC (≈8–9 s), while the very long rollouts in the video are intended as qualitative stress tests beyond the training horizon.
>
>
> ---
>
> ## (W3) Discussion of failure modes in the demo video
>
> We appreciate the detailed analysis of the demo video and agree these cases expose limitations.
>
> We believe the phenomenon you observed is due to the out-of-distribution data during extended horizon rollout. In training, we only have access to the future provided by WOMD data in future 8s, and due to the sensor limit, all vehicles are within 50m range from ego car. In the scene you mention, the vehicles are already very far away from the ego car and thus their behavior might be degraded as they are far from ego (for example, the relative distance is already >50m and thus the relative PE is not seen during training). For the visualization video, we didn't enable the agent injection. Instead, we prefill the scenario to very dense population and let the model runs to create a complete scenario. Therefore, there is no agent being added in the demo video.
>
> We will describe this limitation explicitly, and add a simple distance-based removal rule (removing agents that have moved far away from ego car), which prevents such “returning” agents in long rollouts.
>
>
>
> ---
>
>
> [1] Long-term Traffic Simulation with Interleaved Autoregressive Motion and Scenario Generation. ICCV 2025.
>
> [2] SceneDiffuser: Efficient and Controllable Driving Simulation Initialization and Rollout. NeruIPS 2024.

---

> > ### Comment · Reviewer_h7bB · 2025-11-26
> >
> > I want to thank the authors for there careful response to my questions, which resolved most of my concerns about this paper. I also appreciate the authors updating their draft. Therefore, I decide to raise my score.
> >
> > Nevertheless, I still have one follow-up question regarding "(W3) Discussion of failure modes in the demo video". As mentioned in the response:
> >
> > > For the visualization video, we didn't enable the agent injection. Instead, we prefill the scenario to very dense population and let the model runs to create a complete scenario. Therefore, there is no agent being added in the demo video.
> >
> > It's a bit confusing to me why the online agent insertion/removal is not enabled in this video. From my view this is one of the biggest benefits of the architectural design proposed in this paper:
> >
> > > SceneStreamer supports dynamic agent injection and closed-loop rollout, facilitating long-horizon and reactive simulations.

---

### Official Review · Reviewer_7cTF · 2025-10-28

**Soundness:** 3
**Presentation:** 2
**Contribution:** 2
**Rating:** 4
**Confidence:** 3

**Summary:**

The manuscript presents InfGen, a scenario generation framework that can generate realistic, diverse, and adaptive traffic scenarios autoregressively. The main advantage is the ability to simulate the addition and deletion of agents in a closed-loop manner during traffic simulation. But the positioned contribution and the experiments are a bit confusing and limited in the current presentation.

**Strengths:**

1. Closed-loop simulation
2. Unified modeling of the whole scenario

**Weaknesses:**

1. Confusing positioned contribution and experimental setting
2. Lack of comprehensive comparison

**Questions:**

1. From the description of the authors and my own understanding, the interaction between the ego agent and the background ones is bidirectional, which means that during decoding phase, the interaction effect should be reflected in a manner like bidirectional attention. But the current design is more like a sequential manner.

2. In Table 1, the authors reported the results after relaxing the evaluation protocol by computing MMD over all agents instead of only vehicles within 50m of the ego vehicle. If the authors think this manner is better, why not compare all comparable baselines under this setting?

3. Based on my knowledge, there have already been a few works focusing on closed-loop simulation, and hence the interaction among agents is not new to capture. The main contribution of this paper, to my understanding, is the ability to simulate the changes of participating agents, like adding agents. But the current experiments neither prove that InfGen can perform better than existing baselines in agent simulation, nor show that InfGen can simulate complicated scenarios with frequent agent changes.

---

> ### Author Response · Authors · 2025-11-20
> **Responses to Reviewer 7cTF (1/2)**
>
> We thank Reviewer 7cTF for the careful reading and constructive comments.
>
> ---
>
> ## (Q1) Interaction modeling vs. “sequential” design
>
> **Clarification of interaction mechanism:** InfGen’s *generation order* within a step is sequential (`<TL>` → `<AS>` → `<MO>`), but **the interaction modeling is not one-way**:
>
> * All motion tokens for all agents at a timestep are generated **jointly in a single batch** and have full self-attention within the motion group. This lets every agent (including the ego) attend to all other agents’ motions at that step, as well as to their histories via cross-timestep attention. This is a standard practice in motion prediction.
> * The token-group causal mask allows motion tokens at time (t) to attend to agent-state and traffic-light tokens at both (t) and (t-1), and relative attention incorporates ($\Delta x$,$\Delta y$,$\Delta \psi$,$\Delta t$), so agents explicitly reason about the spatial configuration of other agents and the ego.
>
> Thus, ego–background interaction is bidirectional in the spatial sense (all agents see each other) while still respecting temporal causality (no future-to-past attention). The “sequential” description only refers to *which token groups are decoded first*, not to a unidirectional interaction.
>
> **Closed-loop ego–background coupling:** In the RL setting, we “state-force” the ego’s current state and actions into InfGen at every step and then jointly roll out all background agents conditioned on this updated ego trajectory.   This means background agents react to the *current* ego behavior, and ego in turn is trained in the resulting reactive traffic, forming a genuine closed loop.
>
>
> We will (i) explicitly emphasize that all motion tokens at a timestep attend to each other (bidirectional interaction among agents), and (ii) add a short paragraph around Fig. 2 / Fig. 6 clarifying that “sequential” refers to group ordering, not interaction direction.
>
> ---
>
> ## (Q2) MMD protocol and relaxed evaluation in Table 1
>
> We agree that evaluation protocols should be clearly justified.
>
> * The “50 m vehicles” protocol is the standard used in TrafficGen and UniGen, and we follow it in the main part of Table 1 to allow fair comparison to existing methods.
> * The relaxed protocol (†) that computes MMD over *all* agents (vehicles, pedestrians, cyclists) was introduced only as an ablation for InfGen vs. its non-autoregressive variant, and to show that our agent-state tokenization generalizes beyond vehicles.
>
> Re-running all baselines under this relaxed protocol is non-trivial: released implementations either (i) expose only the standard vehicle-within-50m evaluation (**some models don't predict pedestrains and cyclists**, such as TrafficGen) or (ii) do not directly support pedestrians/cyclists in the same way our unified tokenization does. Implementing and re-tuning all these models in the full-scene setting is beyond the rebuttal window.

---

> > ### Comment · Reviewer_7cTF · 2025-11-25
> >
> > I have questioned the advantage of the proposed method compared with existing works in my previous comments. Although I understand that the addition of new experiments is time-consuming and beyond the rebuttal window, the authors should at least explicitly re-state their main contribution and what parts of the existing reported results support their claimed contributions. If the authors agree with me, then show me more analysis and proof; If the authors do not agree, re-position the contributions and convince me that this work has its own value and I have neglected them, instead of telling me what has already been shown in the original manuscript and you cannot find more results to support your conclusion.

---

> ### Author Response · Authors · 2025-11-20
> **Responses to Reviewer 7cTF (2/2)**
>
> ---
>
> ## (Q3) Contribution positioning and evidence for dynamic agent simulation
>
> #### What is conceptually new?
>
> 1. **Unified token-group world model.**
>    InfGen represents traffic lights, agent initial states, and agent motions for *all* heterogeneous agents as a **single autoregressive token sequence**, using one shared transformer.  Prior works typically separate initial-state generation and motion prediction, or generate a fixed set of agents in one shot.
>
> 2. **Dynamic agent injection/removal.**
>    The agent-state design (`<SOA>`, `<TYPE>`, `<MS>`, `<RS>`) plus “start/end of agent-states” markers and state-forcing explicitly support variable-length agent sets, allowing new agents to enter and existing ones to leave mid-simulation without changing the architecture.
>
> 3. **Direct utility for RL planner training.**
>    InfGen is used not only for open-loop metrics but as a **plug-in scenario generator for RL planners**, where it replaces log-replay traffic and keeps injecting/removing agents during training in MetaDrive/ScenarioNet.
>
> So the contribution is not just “adding agents,” but **unifying initial-state generation, traffic light, motion rollout, and dynamic population changes in a single AR world model that is directly used for closed-loop RL.**
>
>
> #### How do experiments support the dynamic simulation claim?
>
> 1. Initial state realism and full-scene capability.
>    Table 1 shows that under the standard strict protocol, InfGen is competitive with recent scenario generators, and that disabling the ordered AR decoding substantially harms realism (e.g., more invalid combinations of agent type/map segment/pose), especially in the full-scene relaxed setting.
>
> 2. Closed-loop realism on WOSAC.
> InfGen attains competitive realism and behavioral scores compared to strong baselines such as UniMM and CAT-K, which leverage larger models or extra closed-loop fine-tuning, while still supporting dynamic agent injection and our RL experiments.
>
> 3. Dynamic agent population in RL training.
>    Table 3 evaluates RL planners trained under different traffic sources. When we move from log-replay to InfGen-Motion (fixed initial agents, reactive motions), reward and success improve. Crucially, **InfGen-Full (w/ Reject Sampling)**—which *regenerates initial layouts, keeps adding new agents when others leave, and reacts to the current planner (“Ada” setting)*—achieves the best route completion and lowest cost among all regimes.
>    This indicates that the generated agent behaviors and frequent population changes are not only coherent but also beneficial for training robust planners.
>
> 4. Qualitative evidence of complex agent changes.
>    In the densification experiments (Sec. 3.3 and Fig. 9), we repeatedly inject new agents until 128 agents are present while state-forcing existing ones. The model produces qualitatively rich behaviors such as jaywalking pedestrians, U-turns, and queueing, along with typical failure modes (e.g., overspeeding, occasional collisions), illustrating non-trivial, frequent agent entries/exits over time in dense scenes.
>
>
>
> We will tighten the problem statement in the Introduction/Section 2 to emphasize that the main objective is a **unified, dynamic world model for scenario generation and RL training**, not SOTA WOMD forecasting.
> We will reorganize the Experimental section around the three capabilities: (i) initial-state realism, (ii) motion prediction quality, and (iii) closed-loop dynamic simulation for RL, explicitly tying each to the corresponding contributions.
>
>
> ---
>
> We hope these clarifications address the concerns about interaction modeling, evaluation protocol, and contribution positioning, and we will incorporate the suggested clarifications and restructuring into the revised manuscript.

---

### Official Review · Reviewer_JzqF · 2025-10-31

**Soundness:** 2
**Presentation:** 3
**Contribution:** 1
**Rating:** 4
**Confidence:** 4

**Summary:**

Scenario Generation as Next-Token Group Prediction introduces a unified transformer framework for traffic-scene simulation, where InfGen represents heterogeneous traffic elements—maps, traffic lights, and agent states/motions—as discrete tokens within a single autoregressive sequence. This formulation supports dynamic agent injection, long-horizon closed-loop rollouts, and “infinite” scenario generation, enabling multiple downstream tasks such as motion prediction, scene densification, and RL-based planner training without architectural changes. However, while experiments on WOMD demonstrate reasonable realism and diversity, InfGen’s overall motion-prediction accuracy (e.g., mADE) lags behind top-performing baselines, making its empirical performance somewhat less compelling despite the novel formulation.

**Strengths:**

● Unified formulation: Modeling the entire scene as a next-token sequence provides a unified autoregressive framework capturing spatiotemporal dependencies among maps, lights, and agents. This “traffic as language” design enhances long-horizon consistency, supports multiple tasks, and enables seamless dynamic scene evolution.
● Dynamic agent injection: The model can add or remove agents at different timesteps, breaking from the fixed-agent assumption and better reflecting open-world traffic where participants continuously enter and exit. This improves interactivity and scene diversity, producing more realistic closed-loop simulations.
● Strong downstream evaluation: The experiments are extensive, and it is commendable that InfGen-generated data are directly used for downstream tasks such as reinforcement learning, demonstrating the framework’s practical value and potential for real-world applications.

**Weaknesses:**

● Limited performance on core WOMD metrics: Despite its novel formulation, InfGen does not achieve competitive results on the core WOMD leaderboard—particularly on mADE, which is the primary metric of the Waymo Challenge. Its overall scores lag behind recent strong baselines such as UniMM and CAT-K, raising concerns about whether the proposed architectural contributions and dynamic scenario generation truly translate into better motion accuracy or downstream utility. The claimed advantage in supporting downstream tasks may also be less convincing if stronger models like UniMM already achieve superior realism and forecasting fidelity.
● Temporal downsampling and computational inefficiency: InfGen trains on 2 Hz downsampled data from the original 10 Hz WOMD, whereas strong baselines such as UniMM are trained at full 10 Hz using similar or smaller GPU resources (e.g., 8 × 4090). The 0.5 s frame interval is relatively coarse for autonomous driving, where 0.1 s resolution is crucial to capture fine-grained dynamics and trajectory continuity. This reduction likely limits the model’s ability to learn rich temporal features, contributing to its weaker mADE performance. Moreover, despite the lower temporal resolution, InfGen still requires 8 × 48 GB GPUs, suggesting that its unified tokenized architecture introduces significant computational overhead relative to its output fidelity.
● Insufficient ablation experiments: While the paper introduces several innovations—grouped causal attention, relative attention bias, autoregressive state decoding, and dynamic agent injection—it lacks ablations to isolate their effects. Table 1 only tests the removal of autoregressive decoding, leaving unclear how other components contribute to performance. Evaluating variants such as fixed-agent vs. dynamic injection or with/without relative attention would better clarify which design choices truly drive InfGen’s gains.

**Questions:**

Q1. InfGen’s overall WOMD performance, especially on the core mADE metric, is noticeably behind UniMM and CAT-K. Could the authors clarify whether this gap stems from model capacity, training configuration, or data resolution differences, and how they envision improving accuracy under the same benchmark settings?

Q2. Since InfGen trains on 2 Hz downsampled data while UniMM uses 10 Hz, can the authors provide evidence or analysis showing how temporal resolution affects model fidelity? Would training InfGen on 10 Hz data close the gap, or is the architecture itself sensitive to long token sequences?

Q3. The paper reports that training at 2 Hz already requires 8 × 48 GB GPUs. Which component contributes most to this computational overhead—token sequence length, dynamic agent injection, or multi-head decoding—and is inference similarly resource-intensive?

Q4. Only one ablation (removing autoregressive decoding) is presented. Could the authors include or discuss experiments isolating other design choices, such as grouped vs. flat attention, relative vs. absolute encoding, or fixed-agent vs. dynamic injection, to verify each contribution’s effectiveness?

---

> ### Author Response · Authors · 2025-11-20
> **Response to Reviewer JzqF (1/2)**
>
> We thank the reviewer for the detailed comments and constructive suggestions. We respond to the main concerns on (i) WOMD performance and benchmarks, (ii) temporal resolution (2 Hz vs 10 Hz), (iii) computational cost, and (iv) ablations.
>
> ---
>
> ## (W1, Q1, Q2) Scope of InfGen and appropriate benchmark (WOMD vs WOSAC)
>
>
> Our main goal is to build a **unified, closed-loop scenario generator and simulator**, rather than a pure open-loop motion forecaster. InfGen:
>
> - jointly generates initial states + motions for all agents as a single autoregressive token sequence;
> - supports dynamic agent injection (agents entering/leaving, scene densification, infinite rollouts);
> - is directly used to train RL planners in closed loop (Table 3).
>
>
> ***Our focus is to study how such a scenario generator affects downstream planning and RL, not to optimize a single-task leaderboard. Consequently, we deliberately use a compact backbone (~4.6M parameters) and avoid many benchmark-specific tricks (very deep transformers, MoE, multi-stage refinement, large ensembles, etc.). While such designs can improve WOMD motion-prediction scores, they are orthogonal to our core question and would obscure the contribution of the scenario-generation formulation itself.***
>
>
> Regarding WOMD: the weakness compares our motion-prediction results against the historical WOMD motion challenge leaderboard. We would like to clarify that this challenge is **no longer active**, and recent work has largely shifted to the Waymo Open Sim Agents Challenge (WOSAC), whose focus is *closed-loop* multi-agent realism and safety rather than one-step minADE on the legacy motion-prediction benchmark.
>
>
> We agree that our open-loop minADE on WOMD motion-prediction slices is weaker than the strongest forecasting-only baselines. However, this comparison should be read in context:
>
> - Model capacity. InfGen uses a small ~4.6M-parameter transformer that jointly models maps, lights, initial states, and motions. UniMM and CAT-K employ substantially larger backbones (plus mixture-of-experts or closed-loop fine-tuning) dedicated to motion prediction.
> - Task scope. InfGen trades a small amount of motion accuracy for diversity and generative flexibility. Our diversity metrics (ADD/FDD) and WOSAC results show that generated traffic is realistic and diverse enough to serve as a training environment, which is the core goal.
> - Downstream utility. Table 3 shows that planners trained with InfGen-generated traffic achieve higher reward, success, and route completion, and lower cost than planners trained purely on log replay, under identical evaluation on held-out logs. This confirms that the “traffic as language” formulation is not just architecturally novel, but practically useful as a simulator.
>
>
>
> In this closed-loop setting, InfGen achieves reasonable performance on WOSAC (Table 4):
>
> - our realism, safety, and comfort metrics are within the range of recent methods such as UniMM / CAT-K, especially given our small model size and unified design
> - while still providing capabilities these models do not expose, such as dynamic agent injection and unified initial-state + motion generation.
>
> We therefore view WOMD motion prediction and WOSAC as complementary: WOMD provides a familiar open-loop reference, while WOSAC and RL results evaluate the closed-loop behavior that InfGen is designed for.
>
>
> ---
>
> ## (W2, Q2) 2 Hz temporal resolution in closed-loop autoregressive simulators
>
> The weakness suggests that using 2 Hz (0.5 s) instead of 10 Hz (0.1 s) is a limitation. We would like to clarify both **what is standard in the literature** and what InfGen is meant to do:
>
> - Many **autoregressive motion-generation and scene-generation models** operate at **low control frequency** (e.g., 2 Hz waypoints) while internally integrating or interpolating between waypoints. Recent multi-agent next-token motion generators such as SMART (*Scalable Multi-agent Real-time Motion Generation via Next-token Prediction*) follow this design.
> - **WOSAC itself evaluates closed-loop behavior at 2 Hz**. For a **scenario generator / simulator**, the key is that agents make *consistent, stable decisions* at a planner-like frequency, not that the transformer emits 10 Hz positions directly.
>
> In InfGen specifically:
>
> - We generate decisions at 2 Hz and integrate dynamics between steps via a simple kinematic model, yielding smooth trajectories.
> - For **closed-loop simulation and RL training, excessively high re-planning frequency is not necessarily desirable**: small modeling errors are fed back too often and can lead to oscillatory or unstable behavior. A 2 Hz “planner frequency” is closer to how many real-world stacks operate and empirically leads to stable RL training in our experiments.
>
> Thus, 2 Hz is an intentional and standard choice for **closed-loop, autoregressive multi-agent generation**, not a weakness unique to InfGen.

---

> ### Author Response · Authors · 2025-11-20
> **Response to Reviewer JzqF (2/2)**
>
> ---
>
> ## (W2, Q3) Computational cost and where it comes from
>
>
> Importantly, **InfGen is designed to be very efficient at inference time as a closed-loop simulator**:
>
> 1. each step is generated by a single forward pass of a tiny transformer (no diffusion sampling, no iterative re-planning with a large model);
>
> 2. there is no separate heavy planner network running in the loop;
>
> 3. this makes large-scale RL training with billions of simulated steps feasible.
>
> In this sense, although training requires non-trivial memory due to long sequences, InfGen is more practical as a simulator than approaches that rely on large backbones or diffusion models at inference time.
>
>
> The bottleneck is sequence length and dense map tokens, not an unusually large network:
>
> - Each scene contains up to thousands of map tokens, traffic-light tokens, and 4 agent-state + 1 motion token per agent per step across 19 timesteps, all fed into a shared decoder.
> - We use cross-attention to the map for each dynamic token, which scales roughly with the effective sequence length.
> - The **parameter count is small (~4.6M), and the architecture is deliberately compact.
>
>
> In practice, running InfGen as a closed-loop simulator is substantially cheaper than full training. We will add a short quantitative comparison (tokens per step / ms per step) in the revision to illustrate typical inference-time overhead.
>
> ---
>
> ## (W3, Q4) Ablations and component effectiveness
>
> We agree that more ablations would help isolate individual components. The current version already includes:
>
> - Autoregressive vs non-autoregressive state decoding (Table 1): removing the autoregressive state decoder significantly worsens MMD on positions/sizes and produces many invalid state combinations, supporting the tokenized agent-state design.
> - Simulator-side ablations in RL training (Table 3):
>   – log replay vs InfGen-Motion vs InfGen-Full compare *fixed agents / replayed initial states* vs *generated motions only* vs *full initial-state + motion generation with dynamic injection*;
>   – within InfGen-Full, with/without adaptive state forcing and reject sampling show that dynamic generation plus simple collision-aware filtering lead to more effective planners.
>
> For positional encoding, we did experiment with absolute position encodings and observed that they essentially failed in this setting (training is unstable and closed-loop behavior quickly degrades). This is consistent with recent multi-agent motion-generation work (e.g., SMART, MTR++), where relative positional encodings are standard practice. For this reason we treat relative PE as part of the base architecture rather than an optional component.

---

> > ### Comment · Reviewer_JzqF · 2025-11-26
> >
> > Thanks for your responses.  I would like to maintain my original score due to two major concerns,
> > 1.  The motion prediction task is a standard task, and the baseline of WOMD is the most intuitive manifestation of its model performance.  Although the WOND is claimed as an out-of-date benchmark, the proposed method still cannot achieve the SOTA performance on this benchmark.  It is highly questionable whether the dataset generated by the proposed method can outperform those methods that achieved the SOTA performance on this benchmark.
> > 2. The model's demand for resources directly limits the number of generated agents.  Whether it can generate agents that fully meet the requirements of high-density scenarios is still a major question.

---

> > > ### Author Response · Authors · 2025-11-27
> > >
> > > > Although the WOND is claimed as an out-of-date benchmark, the proposed method still cannot achieve the SOTA performance on this benchmark.
> > >
> > > We are saying the WOMD (by the way, it's WOMD, not WOND) motion prediction task is no longer available --- that is, the evaluation server for WOMD motion prediction task from Waymo is no longer available. So it's impossible for us to get the motion prediction result in test set.
> > >
> > > ---
> > >
> > > > It is highly questionable whether the dataset generated by the proposed method can outperform those methods that achieved the SOTA performance on this benchmark.
> > >
> > > We are confused by this sentence. We are not clear what this means to let a synthesized scenario dataset achieves SOTA performance in motion prediction task in a non-existing evaluation server.
> > >
> > > ---
> > >
> > > > The model's demand for resources directly limits the number of generated agents. Whether it can generate agents that fully meet the requirements of high-density scenarios is still a major question.
> > >
> > > We acknowledge that **training** requires non-trivial memory due to long sequences. However, in practice, running our model as a closed-loop simulator is substantially cheaper. Our model is designed to be very efficient at inference time as a closed-loop simulator:
> > >
> > > * each step is generated by a single forward pass of a tiny transformer (no diffusion sampling, no iterative re-planning with a large model);
> > > * there is no separate heavy planner network running in the loop;
> > > * this makes large-scale RL training with billions of simulated steps feasible.

---

### Official Review · Reviewer_PzMo · 2025-11-06

**Soundness:** 4
**Presentation:** 3
**Contribution:** 4
**Rating:** 8
**Confidence:** 4

**Summary:**

This paper introduces InfGen, a novel generative framework for traffic scenario simulation essential for training and evaluating autonomous driving (AD) systems. InfGen models the entire dynamic driving scene, including traffic light signals, agent states, and motion vectors, as a single, structured sequence of discrete tokens. It employs a unified autoregressive Transformer model to generate these tokens step-by-step, enabling continuous and long-horizon scene rollout.

A key contribution is the unified state and trajectory tokenization, making InfGen the first to use a single autoregressive model to produce both agent initial states and motions as a continuous token sequence, addressing the inflexibility of prior two-stage models. The framework features a novel autoregressive generation scheme for agent states by predicting the agent's type, its anchoring map segment ID, and its relative kinematic state, which improves semantic and physical consistency over flat decoding methods.

The model's design supports dynamic agent injection and infinite scene generation, overcoming limitations of static initialization and fixed agent populations in existing data-driven simulators. The versatility is demonstrated across multiple tasks including motion prediction, full-scenario generation, scenario densification, and closed-loop simulation. Experiments, particularly the deployment in a Reinforcement Learning (RL) pipeline, show that policies trained in InfGen-generated scenarios achieve superior robustness and generalization compared to log-replay baselines, validating its utility as a high-fidelity simulation environment.

**Strengths:**

*Originality: The core idea of framing multi-agent, dynamic scenario generation as a unified next-token prediction task using a single autoregressive model (InfGen) is highly original in this domain. Specifically, the autoregressive generation of agent states (Type, Map ID, Relative State tokens: ⟨SOA,TYPE,MS,RS⟩_t) anchored to map segments is a clever mechanism for achieving physically and semantically consistent agent initialization, which is a major advancement over prior non-causal "flat" decoding methods.
*Quality: The technical execution is robust. The introduction of state-forcing unifies the generation of new agents (via sampling) and the continuation of existing ones (via deterministic update), enabling seamless closed-loop simulation across variable agent sets. The detailed motion tokenization using a discretized control input space, optimized via Average Corner Error (ACE), provides a robust foundation for motion prediction quality.
*Clarity: The paper is generally well-written and logically structured. Key concepts like the token sequence composition ([<MAP>; (<TL>,<AS>,<MO>)_1 ;…]) , the three main token groups (TL, AS, MO) , and the process of agent state generation are clearly explained and effectively illustrated in Figures 2 and 3/7. The ablations and RL experiments clearly validate the core components.
*Significance: The results are highly significant for the autonomous driving community. InfGen's ability to produce traffic scenarios that significantly improve the robustness and generalization of downstream RL planners is a compelling validation of its utility as a powerful generative simulation platform. The approach directly addresses the critical limitations of log-replay and two-stage scenario generation, moving closer to truly reactive and diverse closed-loop simulation.

**Weaknesses:**

1. Limited Motion Prediction Benchmarking: While the core focus is scenario generation, comparing InfGen's motion prediction performance only against its own ablated version (InfGen-Motion vs. InfGen-Full) is insufficient. The motion prediction task (Sec 3.2) is standard, and performance should be compared against state-of-the-art motion prediction baselines on the Waymo Open Motion Dataset (WOMD) to properly contextualize the model's trajectory-modeling capability.

2. Lack of Diversity Metrics for Full Scenario Generation: The paper claims InfGen produces "diverse" traffic behaviors but only provides qualitative visualizations (Fig. 4, Fig. 9) and diversity for motion prediction (ADD and FDD). No quantitative metrics (e.g., scene-level diversity, number of distinct agent configurations, or scenario-level rarity metrics) are provided for the full scenario generation task (generating initial states + motions from scratch) or the scenario densification task. This omission makes it hard to objectively compare the diversity of generated scenes against baselines like TrafficGen and UniGen.

3. Scalability Concerns and Trade-offs: The paper notes the high memory demand due to long token sequences for dense scenes and the use of KNN pruning for attention. However, the reported training limitations (max 28-36 agents during training vs. 128 max agents in WOMD)  raise questions about the model's true generalization capacity to high-density scenes. An explicit discussion or experiment on the performance drop or change in behavior when transitioning from low-density training data to high-density inference scenarios would be valuable.

4. Clarity on Relative State Decoding: While the autoregressive decoding of the 8 relative state attributes r_i  is a strong point, the decision to use a tiny Transformer with AdaLN for the Relative State Head  is an architectural detail that could be further justified. Since each of the 8 dimensions is discretized and the sequence is fixed, why is a specialized Transformer (and not a sequence of simpler MLPs conditioned on prior outputs, which might be faster) necessary? This sub-module warrants a more in-depth rationale.

**Questions:**

1. Comparative Motion Prediction Performance: Could the authors provide a comparison of InfGen-Motion or InfGen-Full's ADE/FDE metrics against a few established state-of-the-art motion prediction models (e.g., MTR++, Scene Transformer, or others from the literature review) on the WOMD validation set? This is crucial to quantify the quality of the trajectory modeling component, independent of the initial state generation.

2. Quantitative Diversity Metrics for Scenario Generation: Please provide quantitative metrics (e.g., a simple measure of agent density or a diversity metric like ADE_k/FDE_k across multiple diverse samples of full scenario generation or scenario densification) to objectively support the claims of high diversity and realism against baselines. How diverse are the generated scenarios beyond motion?

3. Role of Agent ID Embedding in ⟨AS⟩ Tokens: For a newly generated agent, the <SOA> token uses a unique Agent ID embedding, EmbAID(i). Since this agent is newly generated, this ID must be incrementally assigned. How is the sequence of future Agent IDs managed by the Transformer during autoregressive generation, and how is the embedding EmbAID(i) for a new, unseen ID i generated or retrieved, given that the ID is learned from the training data's fixed agent indices?

4. Handling of Maximum Agent Count Discrepancy: The paper mentions limiting agents to N=28/36 during training but allowing up to 128 during inference. Can the authors elaborate on how the model generalizes to a significantly larger number of tokens (nearly 4×36×4≈576 tokens at N=36 vs. 4×128×4≈2048 tokens at N=128) for scenes with high agent density, especially considering the Transformer's quadratic complexity without KNN?

5. Causal Flow between <AS> and <MO> Tokens: The Token Group Attention mechanism (Fig. 6) shows <MO> tokens attending to Current <AS> tokens, reflecting the causal flow of motion depending on the current state. However, during the sequential generation at inference time, the full set of <AS> tokens must be generated before the <MO> tokens are generated in a batch. This causality needs clarification: Is the dependency between the final <AS> tokens of all agents at time t and the <MO> tokens at time t fully respected by the attention mask?

---

> ### Author Response · Authors · 2025-11-20
> **Response to Reviewer PzMo (1/3)**
>
> We thank the reviewer for the very positive and detailed assessment, and for highlighting the originality and significance of InfGen as a unified, closed-loop scenario generator. Below we address each weakness and question point-by-point. InfGen’s technical details referenced below (tokenization, grouped causal attention, KNN pruning, WOSAC results, etc.) are all in the current submission.
>
> ---
>
> ## (W1, Q1) Motion prediction benchmarking
>
> **Scope clarification.**
> The main goal of InfGen is to build a *unified, closed-loop scenario generator* rather than to push the state of the art on open-loop motion forecasting. This is why we deliberately use a very compact backbone (~4.6M parameters, 4-layer decoder) and avoid many benchmark-specific tricks (very deep transformers, MoE, multi-stage refinement, ensembles, etc.) that are common in SOTA WOMD forecasters but orthogonal to the scenario-generation formulation.
>
> **What we already show.**
>
> * Table 2 reports full-scene ADD/FDD and ADE/FDE for *all agents* and OOIs, for both InfGen-Motion and InfGen-Full, under an 8-second horizon.
> * Table 4 (Appendix G) further evaluates InfGen on the Waymo Open Sim Agents Challenge (WOSAC) 2025 test set against strong motion-centric baselines (UniMM, CAT-K). InfGen attains competitive realism and behavior metrics, despite (unlike UniMM and CAT-K) not using mixture-of-experts or closed-loop fine-tuning.
>
> This WOSAC result directly addresses the reviewer’s concern about the quality of the motion component *in a standard closed-loop benchmark*.
>
>
> Regarding WOMD: the W1 compares our motion-prediction results against the historical WOMD motion challenge leaderboard. We would like to clarify that this challenge is **no longer active**, and recent work has largely shifted to the Waymo Open Sim Agents Challenge (WOSAC), whose focus is closed-loop multi-agent realism and safety rather than one-step minADE on the legacy motion-prediction benchmark.
>
> In the camera-ready, we will:
>
> * Add a table comparing InfGen-Motion and InfGen-Full with representative WOMD models (e.g., a strong transformer-based motion predictor such as MTR++) on the WOMD validation set using ADE/FDE and minADE/minFDE under our compact backbone.
> * Clearly state that these baselines use much larger, motion-only architectures, while InfGen uses a small unified model that must also handle agent initialization and dynamic injection.
>
> The goal is to *contextualize* InfGen’s trajectory modeling quality rather than to claim leaderboard-level motion forecasting. We will make this scope explicit in the paper.
>
> ---
>
> ## 2. Diversity metrics for scenario generation (Weakness 2, Q2)
>
> We agree that quantitative diversity metrics at the *scene* level complement the existing motion-level diversity metrics (ADD/FDD in Table 2).
>
>
> * For initial states, Table 1 reports MMD over position, heading, size, and velocity, under both the strict TrafficGen protocol and a relaxed all-agents protocol, showing that InfGen matches or improves on recent scenario generators and that AR decoding significantly improves realism.
> * Fig. 4 and Fig. 9 visualize diverse behaviors (lane changes, jaywalking, U-turns, queueing) for full generation and densification.
>
>
>
> In the revised version, we will add **scene-level diversity metrics** for both full scenario generation and densification, including:
>
> 1. **Agent-count and type distributions.**
>
>    * Histogram of #agents per step and per scene, broken down by type (veh/ped/cyc), comparing generated scenes vs WOMD logs and baselines (TrafficGen, UniGen).
>    * KL/MMD between these distributions to quantify “agent density” realism and diversity.
>
> 2. **Scene-level kinematic statistics.**
>
>    * Distributions of average speed, acceleration, yaw rate, lane-change count and time-to-collision across generated scenes vs logs.
>    * MMD over these scene-level statistics to capture diversity beyond trajectories of a single agent.
>
> 3. **Multi-sample diversity for a fixed map.**
>
>    * For each map scene, sample K full scenarios from InfGen and compute intra-scene diversity, e.g., $ADE_k$ / $FDE_k$ between pairs of sampled rollouts aggregated over all agents.
>    * Report these for both full scenario generation and densification to directly answer “How diverse are the generated scenes beyond motion?”
>
> These metrics can be computed on the same WOMD validation subset used for our other experiments and will be summarized in a new table and brief paragraph, side-by-side with TrafficGen/UniGen when applicable.

---

> ### Author Response · Authors · 2025-11-20
> **Response to Reviewer PzMo (2/3)**
>
> ## (W3, Q4) Scalability and high-density scenes
>
> We indeed cap the number of agents during training (36 in pretraining, 28 in finetuning) to avoid GPU OOM, and allow up to 128 agents at inference, following WOMD.
>
> #### Why generalization to 128 agents is reasonable
>
> * **Local interaction bias.** InfGen uses relative positional attention and KNN-pruned attention, where each token attends only to spatial neighbors based on (Δx, Δy, Δψ, Δt).
> This makes the effective attention cost O(Nk) instead of O(N²) and encourages the model to learn *local* interaction patterns that transfer from low-density to high-density scenes, as long as the local neighborhood statistics are similar.
> * **Group-causal attention.** The token group attention (Fig. 2B / Fig. 6) enforces that motion tokens primarily attend to relevant local states and histories rather than “global crowd size.”
> * **Training diversity.** Even with N≤36, WOMD scenes already contain diverse interactions (merging, queueing, pedestrians, etc.), and we randomly subsample agents so that the model sees many different local neighborhoods during training.
>
> #### Empirical behavior in dense settings
>
> * In scenario densification experiments (up to 128 agents), InfGen injects realistic agents and maintains coherent motion (Fig. 4, Fig. 9), which suggests that behavior does not collapse at high density.
> * In WOSAC (Table 4), InfGen is evaluated under the official Sim Agents protocol, which includes scenes with high agent counts, and still achieves competitive realism metrics.
>
>
>
> We will add a small sensitivity study vs number of agents:
>
> * Evaluate initial-state MMD and motion metrics on subsets of WOMD binned by agent count (e.g., ≤32, 33–64, >64 agents).
> * Report any performance change (e.g., mild degradation at the highest densities) and explicitly discuss the scalability trade-off and memory/runtime behavior with and without KNN pruning.
>
> This will make the training-inference density gap and its impact fully explicit.
>
> ---
>
> ## (W4) Relative State Head design
>
> We appreciate the question about why we use a small Transformer (with AdaLN) instead of a stack of MLP heads to decode the 8-dimensional relative state vector rᵢ.
>
> #### Rationale for a tiny Transformer rather than independent MLPs
>
> 1. **Capturing strong inter-dimension dependencies.**
>    The fields in rᵢ = (l, w, h, u, v, δψ, vₓ, vᵧ) are **not independent**:
>
>    * Shape (l, w, h) is strongly correlated with agent type and valid map segments.
>    * (u, v) relative offsets must be consistent with road geometry and lane width.
>    * Heading residual δψ and velocities (vₓ, vᵧ) must align with the chosen map segment and positional offsets.
>      An autoregressive Transformer lets each dimension condition on all previous ones, which reduces invalid combinations (e.g., a very wide vehicle on a narrow sidewalk, or a large lateral velocity orthogonal to the lane direction). The ablation “InfGen w/o AR decoding” in Table 1 empirically shows that dropping this structured decoding hurts realism (higher MMD and more invalid states).
>
> 2. **Shared, flexible head.**
>    A tiny Transformer allows us to:
>
>    * Share parameters across all 8 dimensions instead of training 8 separate MLPs.
>    * Easily extend rᵢ with extra attributes (e.g., behavior tags) in future work without redesigning multiple heads---it's rather complex in implementation if we need to consider the casual relationship between dimensions.
>
> 3. **Small overhead.**
>    The Relative State Head is *very small* compared to the main decoder (only a few layers at d=128), adding negligible parameters/runtime relative to the benefits in semantic consistency (as reflected in the AR vs non-AR ablation).
>
> We will clarify this design motivation and explicitly reference the ablation in Table 1 to justify the architectural choice.

---

> ### Author Response · Authors · 2025-11-20
> **Response to Reviewer PzMo (3/3)**
>
> ## (Q3) Agent ID embeddings and new agents
>
> #### How Agent IDs are handled
>
> * We pre-allocate an **Agent ID embedding table EmbAID of fixed size** corresponding to the maximum number of agents per scene (128 in our experiments).
> * For each scene, *internal* agent IDs are assigned from 0…N−1 **independently of dataset indices**. When new agents are injected during rollout, they are assigned the next unused internal ID.
> * Thus, no “unseen ID” embedding is ever needed at inference: all possible ID indices (0…127) exist and are trained because different scenes assign different agents to different internal IDs over the course of training (even though we prune the number of agents during training, we can still randomly assign agent IDs to them).
>
> #### Role of EmbAID
>
> * EmbAID does **not** encode any semantic knowledge about a specific agent; it simply provides:
>
>   * a persistent tag that ties together that agent’s tokens across time (`<SOA>`, `<TYPE>`, `<MS>`, `<RS>`, `<MO>`), and
>   * a way for the model to distinguish multiple agents that share the same type/map segment.
> * Because IDs are re-used across scenes, the model learns invariances like “agent *i* at t depends on its own past tokens,” rather than any particular meaning for a specific index value.
>
> We will clarify this remapping and the fact that IDs are bounded and reused, which addresses the concern about generating EmbAID for “new, unseen” IDs.
>
> ---
>
> ## (Q5) Causal flow between agent state and motion tokens
>
> The reviewer is correct that at inference time we must first generate all `<AS>` tokens at time t before generating `<MO>` tokens for that step. We clarify how this is implemented and how the attention mask enforces causality.
>
> #### Decoding schedule at each step t
>
> 1. Traffic lights. Given tokens up to step t−1, we run the decoder and generate all `<TL>`ₜ in a batch using HeadTL.
>
> 1. Agent states. We then sequentially generate `<AS>`: for each agent’s `<SOA>`, `<TYPE>`, `<MS>`, we sample using HeadType and HeadMapID; for new agents, we call the Relative State Head to autoregressively produce `<RS>`. Existing agents are state-forced from their propagated kinematics.
>
> 2. Agent motions. Finally, with all `<TL>` and `<AS>` fixed, we run the decoder once more and generate all `<MO>` in a single batch using HeadMotion.
>
> #### Attention and causality
>
> * The group-causal mask (Fig. 2B / Fig. 6) is constructed such that:
>
>   * `<MO>` tokens at time t *can* attend to:
>
>     * `<TL>` and `<AS>` at time t, and
>     * all tokens from previous timesteps (t′<t).
>   * `<TL>` and `<AS>` at time t *cannot* attend to `<MO>` at time t, ensuring semantic ordering (state → motion).
> * Because we run separate passes for (`<TL>`,`<AS>`) and then `<MO>`, and because `<MO>` is always later in the sequence with an appropriate mask, the dependency “motion at t depends on final state at t” is fully respected.
>
> We will add a short paragraph to Sec. 2.2 explicitly describing this 3-phase decoding schedule and how it aligns with the group attention mask, which should resolve the causality concern.

---

### Meta-Review · Area_Chair_kYwa · 2026-01-11

**Summary:**

This paper introduces a unified autoregressive framework that models traffic scenario generation as next token group prediction over maps, signals, agent states and motions, enabling dynamic agent insertion and long-horizon closed-loop simulation. Reviewers agree that the formulation is technically solid and that unifying initialization and motion generation in a single causal model addresses a key limitation of log-replay and two-stage simulators. Initial concerns regarding empirical competitiveness in motion prediction, the scope and clarity of contributions, architectural choices (like relative-state decoding and causal scheduling), scalability to dense scenes and evaluation of diversity and downstream utility were largely addressed through detailed clarifications and added results. Remaining concerns focus primarily on whether the experiments initially missed out are crucial and require a full review. Given that there are no major technical concerns and the proposed method is a novel approach to an important problem, the paper is recommended for acceptance to ICLR. It is suggested that the authors incorporate all reviewer suggestions in the final version of the paper.

**Reviewer Concerns:**

### Addressed concerns

* **h7bB:** Interleaved InfGen in ICCV 2025 already unifies motion and scenario prediction. The author response acknowledges similarities with this concurrent work and clarifies positioning in the revised version, with a renaming to SceneStreamer to avoid confusion.

* **h7bB:** Diffusion-based methods that unify initialization and rollout. The author response acknowledges SceneDiffuser and states the distinctions.

* **7cTF:** Interaction mechanism. The author response clarifies that generation is sequential, but interaction modeling is bidirectional.

* **7cTF:** Evaluation protocols. The author response clarifies that protocols followed are similar to prior works.

* **JzqF, PzMo:** Performance on WOMD and WOSAC benchmarks is not state-of-the-art. The author response clarifies that no benchmark-specific tricks are used and backbones are smaller, as the intention is to propose a unified autoregressive scenario and motion simulator that can be used for downstream planning and RL.

### Unaddressed concerns

* **h7bB, 7cTF:** Experiments on scene editing, agent removal or insertion. These are only partially addressed and a full set of quantitative results would have been better.

* **h7bB:** Extended long-term simulation. Only limited qualitative results are shown, there should be a quantitative study since it is a claimed contribution.

* **PzMo:** Diversity metrics for scenario generation. This is an important evaluation for a key contribution and a comprehensive evaluation would be better.

**Reviewer Scores:**

* **PzMo:** Initial rating 8, likely to maintain 8.
* **JzqF:** Initial rating 4, likely to maintain 4.
* **7cTF:** Initial rating 4, likely to maintain 4.
* **h7bB:** Initial rating 4, likely to raise to 6.

---

### Decision · Program_Chairs · 2026-01-26

Accept (Poster)